# Biosynthesis of histone messenger RNA employs a specific 3′ end endonuclease

Ilaria Pettinati[1], Pawel Grzechnik[2], Claudia Ribeiro de Almeida[3], Jurgen Brem[1], Michael A McDonough[1], Somdutta Dhir[3], Nick J Proudfoot[3]*, Christopher J Schofield[1]*

[1]Department of Chemistry, University of Oxford, Oxford, United Kingdom; [2]School of Biosciences, University of Birmingham, Birmingham, United Kingdom; [3]Sir William Dunn School of Pathology, University of Oxford, Oxford, United Kingdom

**Abstract** Replication-dependent (RD) core histone mRNA produced during S-phase is the only known metazoan protein-coding mRNA presenting a 3′ stem-loop instead of the otherwise universal polyA tail. A metallo β-lactamase (MBL) fold enzyme, cleavage and polyadenylation specificity factor 73 (CPSF73), is proposed to be the sole endonuclease responsible for 3′ end processing of both mRNA classes. We report cellular, genetic, biochemical, substrate selectivity, and crystallographic studies providing evidence that an additional endoribonuclease, MBL domain containing protein 1 (MBLAC1), is selective for 3′ processing of RD histone pre-mRNA during the S-phase of the cell cycle. Depletion of MBLAC1 in cells significantly affects cell cycle progression thus identifying MBLAC1 as a new type of S-phase-specific cancer target.
DOI: https://doi.org/10.7554/eLife.39865.001

## Introduction

During S-phase of the cell cycle, production of the core histone proteins (H2A, H2B, H3, and H4) is coordinated with DNA replication (*Harris et al., 1991*; *Ewen, 2000*). Metazoan mRNAs encoding for the 'replication-dependent' (RD) core histones lack the normal polyA tail formed by 3′ end hydrolysis of pre-mRNA followed by polyadenylation (*Proudfoot, 2011*). Instead, they undergo endonucleolytic cleavage at the 3′ side of an RNA hairpin (stem loop) producing mRNA with a 3′stem loop (SL), which is exported from the nucleus for use in translation (*Marzluff et al., 2008*; *Marzluff and Koreski, 2017*). By contrast, the pre-mRNA of replication-independent histone variants are normally polyadenylated and constitutively expressed during the cell cycle (*Marzluff et al., 2002*; *Wagner et al., 2007*).

A single endonuclease, cleavage and polyadenylation specificity factor 73 (CPSF73), is proposed to be responsible for the hydrolysis of both RD histone pre-mRNA (SL) and normal protein-coding pre-mRNA (polyA) (*Dominski et al., 2005*; *Mandel et al., 2006*; *Kolev et al., 2008*; *Sullivan et al., 2009b*; *Dominski, 2010*). Although maturation of both classes of RNAs requires a hydrolytic reaction, different macromolecular complexes are recruited to the different pre-mRNA classes (SL or polyA) (*Kolev et al., 2008*; *Sullivan et al., 2009b*). Specific factors involved in RD histone pre-mRNA maturation are the stem loop binding protein (SLBP), the FLICE-associated huge protein (FLASH), and the U7 small nuclear ribonucleoprotein (U7snRNP) that binds to a histone downstream element (HDE) (*Dominski et al., 2005*; *Kolev et al., 2008*; *Sullivan et al., 2009b*). Defective 3′ end processing of RD histone pre-mRNA, caused by depletion of factors including CPSF73, SLBP, Sm-like protein 11 (Lsm11), CstF64 or FLASH, results in the generation of extended transcripts downstream of the HDE sequence (*Sullivan et al., 2009a*; *Romeo et al., 2014*; *Sullivan et al., 2009b*). In *Drosophila melanogaster* and humans, misprocessed RD histone pre-mRNA has been observed to undergo polyadenylation involving utilization of a secondary polyadenylation signal sequence located

*For correspondence:
nicholas.proudfoot@path.ox.ac.uk (NJP);
christopher.schofield@chem.ox.ac.uk (CJS)

downstream of the HDE (*Sullivan et al., 2009b*; *Romeo et al., 2014*; *Kari et al., 2013*). Depletion of factors belonging to the 5' cap-binding complex (CBC) (*Hallais et al., 2013*, *Narita et al., 2007*; *Gruber et al., 2012*), or to the cleavage factor II (CF IIm), which is normally involved in 3' end processing of normal protein-coding pre-mRNA (polyA) (*Hallais et al., 2013*; *de Vries et al., 2000*), also results in extended RD histone pre-mRNA transcripts (*Hallais et al., 2013*). These observations suggest a complex and dynamic relationship between the factors involved in the different stages of the RD histone pre-mRNA transcription process, which may involve participation of factors normally belonging to the polyA mRNA processing machinery.

Important cancer medicines, including histone deacetylase and cyclin-dependent kinase inhibitors, target proteins involved in the S-phase (*Newbold et al., 2016*; *Falkenberg and Johnstone, 2014*). In work aimed at identifying potential new S-phase cancer targets, we considered known and potential roles of MBL-fold proteins involved in nucleic acid hydrolysis (*Dominski, 2007*; *Pettinati et al., 2016*; *Daiyasu et al., 2001*). In addition, to the role of CPSF73, and the likely pseudo-enzyme CPSF100, in pre-mRNA processing (*Dominski et al., 2005*; *Mandel et al., 2006*), MBL-fold nucleases are involved in DNA repair (SNM1A-C nucleases) (*Yan et al., 2010*), snRNA processing (INTS9 and INTS11), and tRNA processing (ELAC 1 and 2) (*Skaar et al., 2015*; *Vogel et al., 2005*). Whilst most of the ~18 human MBL-fold proteins have established functions (*Pettinati et al., 2016*), the functions of several are unassigned, including the MBL domain containing protein 1 (MBLAC1). Here, we report evidence that MBLAC1 is a nuclease specific for cleavage of RD histone pre-mRNA. Crystallographic and biochemical studies show that MBLAC1 has an overall MBL fold and di-zinc ion containing active site related to that of CPSF73, but which has distinctive structural features involving active site flanking loops and the absence of the β-CASP domain, which is only present in CPSF73. MBLAC1 depletion from cells leads to the production of unprocessed RD histone pre-mRNA due to inefficient 3' end processing. The consequent depletion of core histone proteins correlates with a cell cycle defect due to a delay in entering/progressing through S-phase.

## Results

### MBLAC1 structure reveals similarity with MBL-fold nucleases

On the basis of sequence similarity studies MBLAC1 has been assigned as an RNAse Z and glyoxalase II subfamily enzyme (*Ridderström et al., 1996*; *Sievers et al., 2011*) (*Figure 1—figure supplement 1A*). However, we found that recombinant MBLAC1 prepared from *E. coli* has only low, likely non-specific, glyoxalase activity as observed for other hMBL-fold proteins belonging to the same subfamily (*Shen et al., 2011*). To investigate its function, we solved a crystal structure of MBLAC1 (1.8 Å resolution, space group P1) (*Table 1*). The structure reveals a stereotypical αββα MBL- fold (*Carfi et al., 1995*) with two central mixed β-sheets (I and II), comprised of 8 and 5 strands respectively, surrounded by helices (*Figure 1A*). In β-sheet I, β-strands 1, 2, 5–6 and 8–10 are anti-parallel, with β-strands 6–8 being parallel; β-strands 3 and 4 are part of a loop region and are aligned anti-parallel to each other and parallel to β-strands 2 and 5, respectively. In β-sheet II, β-strands 11, 12, 13 and 14 are anti-parallel, and β-strands 14 and 15 are parallel (*Figure 1A*). MBLAC1 has four of the five characteristic MBL-metal-binding motifs (*Pettinati et al., 2016*), His116, His118 Asp120 and His121 (motif II), His196 (motif III), Asp221 (motif IV) and His263 (motif V) (*Figure 1B*) (using BBL numbering) (*Galleni et al., 2001*) with two waters completing metal coordination. In the structure of recombinant MBLAC1 produced in *E. coli*, the active site was refined with two iron ions present (*Figure 1B*). However, MBLAC1 produced in HEK293 cells preferentially binds zinc ions (*Figure 1C*). Four MBLAC1 molecules (chains A-D) are present in the crystallographic asymmetric unit; analysis of interactions at the crystallographically observed monomer interfaces (*Krissinel and Henrick, 2007*) identified interactions between chains A-B and C-D (*Figure 1D*) possibly reflecting dimeric MBLAC1 in solution (*Figure 1E and F*). The metal containing active site is adjacent to the dimer interface (*Figure 1D*), rationalizing reduced dimerization as manifested by metal ligand substitution or metal removal (*Figure 1E–G*).

Comparison of the MBLAC1 structure with those of other MBL-fold proteins reveals that it is indeed part of the RNAse Z/glyoxalase II MBL structural subfamily (*Figure 1H*; *Figure 1—figure supplement 1B–C*). However, although there is low overall sequence similarity (27%), the overall MBLAC1 fold is structurally similar to the human endoribonuclease β-lactamase-like-protein 2

**Table 1.** Crystallographic data and refinement statistics PDB ID 4V0H.

| | Native HSE (PDB ID: 4V0H) |
|---|---|
| Data collection | |
| Space group | P1 |
| Cell dimensions | |
| $a, b, c$ (Å) | 62.95, 67.13, 67.90 |
| $\alpha, \beta, \gamma$ (°) | 109.31, 105.40, 90.17 |
| Resolution (Å) | 45.96–1.79 (1.84–1.79 Å)[*] |
| $R_{merge}$[†] | 0.10 (0.81) |
| $I/\sigma(I)$ | 12.4 (2.6) |
| Completeness (%) | 95.5 (92.6) |
| Redundancy | 6.9 (6.7) |
| | |
| Refinement | |
| Resolution (Å) | 45.95–1.79 |
| No. reflections | 90641 |
| $R_{work}$[‡]/$R_{free}$[§] | 0.182/0.211 |
| No. atoms | |
| Protein | 6235 |
| Ligand/ion | 38 |
| Water | 514 |
| $B$ factors | |
| Protein | 27.27 |
| Ligand/ion | 44.40 |
| Water | 33.94 |
| R.m.s. deviations | |
| Bond lengths (Å) | 0.01 |
| Bond angles (°) | 1.37 |

[*] Values in parentheses are for highest-resolution shell.

[†]Rmerge = $\sum_h \sum_l \mid I_{hl} - <I_h> \mid / \sum_h \sum_l <I_h>$ where $I_{hl}$ is the $l$th observation of reflection h, and $<I_h>$ is the mean intensity of that reflection.

[‡]Rwork = $\sum \mid\mid F_{obs}\mid -\mid F_{calc}\mid\mid/\mid F_{obs}\mid$ × 100.

[§]Rfree is calculated in the same way as Rwork but using a test set containing 5.01% of the data, which were excluded from the refinement calculation.

DOI: https://doi.org/10.7554/eLife.39865.004

(*Levy et al., 2016*) (LACTB2), RMSD 2.23 Å over 153 Cα atoms (*Laskowski et al., 2005*) (*Figure 1H*), an endoribonuclease responsible for mitochondrial mRNA maturation (*Levy et al., 2016*). Comparison of the MBLAC1 and LACTB2 active sites reveals very similar di-metal ion binding modes and proximate active site residues (*Figure 1—figure supplement 1B*). Two loops close to the MBLAC1 active site (β3-β4 and β14-α3 loops) are also present in LACTB2 (β1-β2 and β11-α3 loop) (*Figure 1H*), suggesting the enzymes have similar substrate recognition/catalytic mechanisms. Studies on the 'true' MBLs involved in β-lactam antibiotic resistance, imply the two active site loops of MBLAC1 are likely mobile and involved in induced fit binding and in substrate recognition (*Brem et al., 2015*), thus it is possible that the crystallographically observed loop conformations reflect those involved in productive substrate binding/catalysis. Two other regions in the MBLAC1 structure (aa 51–66 and the C-terminal region, aa 239–266) are disordered, implying flexibility and possible involvement in substrate recognition. LACTB2 is reported to have high overall structural similarity with CPSF73 (*Levy et al., 2016*). However, whilst comparison of the MBL domains of

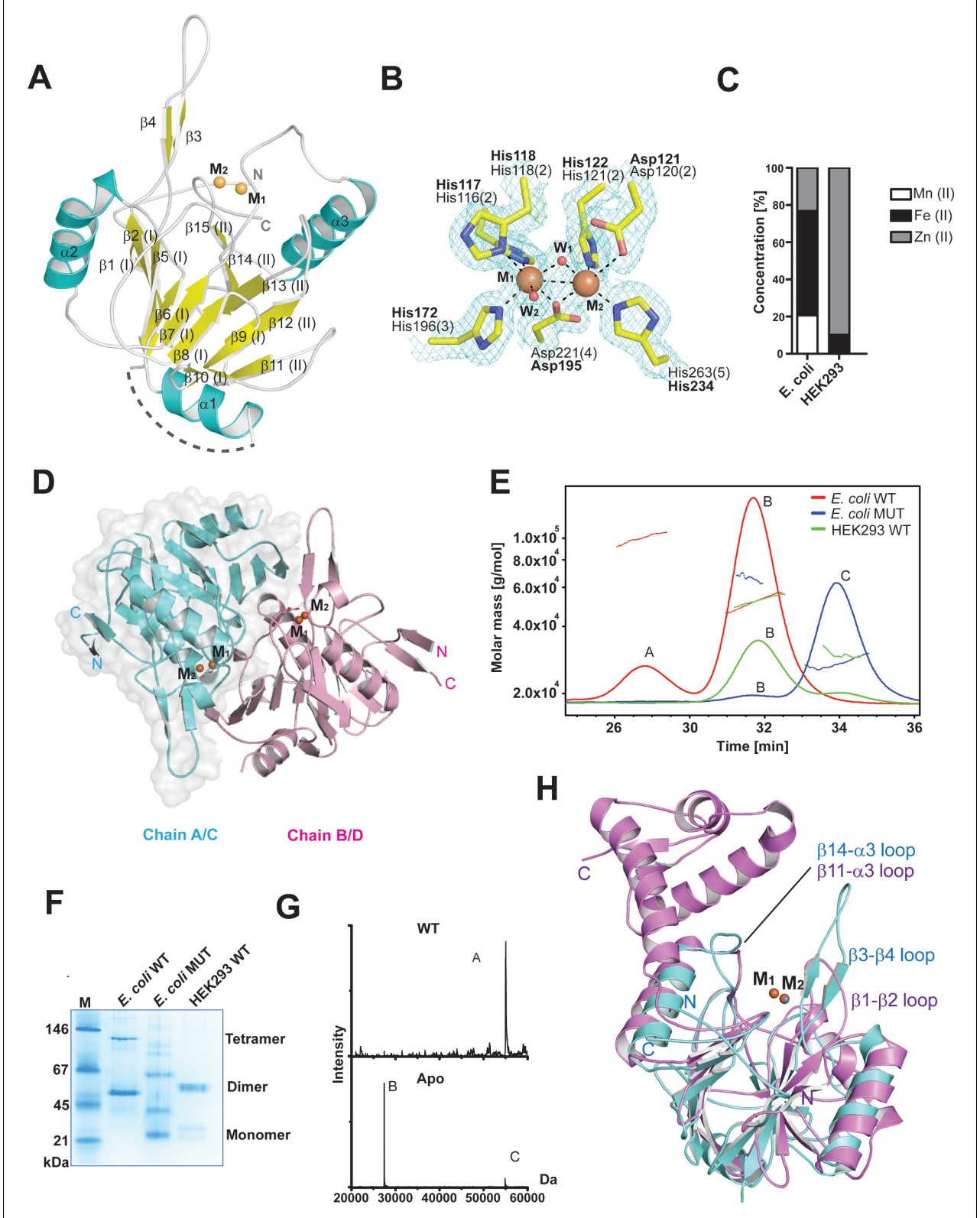

**Figure 1.** MBLAC1 is a dimeric metallo β-lactamase fold protein related in structure to LACTB2. (**A**) View from an MBLAC1 crystal structure (PDB ID: 4V0H) showing secondary structure elements and the di-metal containing active site. Helices are in cyan, β-strands in yellow, and metal ions are orange spheres. Dashed line (in gray) indicates unrefined residues (aa 51–66). (**B**) View of metal binding and active site residues with a representative electron density map (3.0 σ mFo-DFc OMIT; cyan mesh) for the sidechains of His116 (Nε2 to $M_1$: 2.33 Å), His118 (Nε1 to $M_1$: 2.35 Å), Asp120 (Oδ2 to $M_2$: 2.9 Å),

*Figure 1 continued on next page*

*Figure 1 continued*

His121 (Nε2 to M$_2$: 2.37 Å), His196 (Nε2 to M$_1$: 2.27 Å), Asp221 (Oδ2 to M$_1$: 2.14 Å; Oδ2 to M$_2$: 2.12 Å), His263 (Nε2 to M$_2$: 2.06 Å) and the two water molecules (red spheres) which coordinate (black dashed lines) to the metals (orange spheres). The BBL MBL numbering system is used (*Galleni et al., 2001*) (active site motif numbers in parentheses) (*Pettinati et al., 2016*). Numbering as used in our PDB deposited structure is in bold. (C) Inductively coupled plasma mass spectrometry (ICP-MS) experiments reveal that in human cells MBLAC1 binds zinc. (D) MBLAC1 dimer model calculated by PISA (*Krissinel and Henrick, 2007*). A surface representation is shown for protomer A/C. (E) Multi-angle laser light scattering (MALS) analyses of *E. coli* produced wt and active site variant MBLAC1, and of HEK293 produced wtMBLAC1. Peak A likely represents a trimer (~80000 Da); peak B (~54000 Da) a dimer; and peak C (~27000 Da) a monomer. wt MBLAC1 is predominantly dimeric, the active site variant is mainly monomeric. The latter observation correlates with loss of activity of active site variant MBLAC1 (main text *Figure 4a*), the proposal that the dimer is catalytically active, and the observation that the active site is close to the dimer interface. MALS experiments were carried out at the Biophysical Services of the Biochemistry Dept., Oxford University. (F) Non-denaturing PAGE analyses indicates wt MBLAC1 produced in either *E. coli* or HEK293 cells is predominantly dimeric; a higher oligomeric state (tetramer) is also observed for the bacterial produced wt protein. The active site substituted enzyme produced in *E. coli* shows loss of oligomerization. wtMBLAC1 produced in either, *E. coli* or HEK293 cells shows the same oligomerization behavior. (G) Non-denaturing electrospray ionization mass spectrometry deconvoluted spectra of recombinant MBLAC1 produced in *E. coli* indicates that MBLAC1 is dimeric, binding two metal ions. Peak A (54880 Da) represents the dimer with two divalent transition metal ions (zinc or iron) bound to each monomer (+224 Da); peaks B and C (27440 and 54880 Da, respectively) correspond to monomer and dimer without bound metal, following metal removal using EDTA. (H) Superimposition of the MBLAC1 (cyan; metals in orange) and LACTB2 folds (PDB ID: 4AD9) (*Levy et al., 2016*) (pink; metals in grey) reveals a high degree of overall similarity (RMSD 2.2 Å over 153 Cα atoms).
DOI: https://doi.org/10.7554/eLife.39865.002

The following figure supplement is available for figure 1:

**Figure supplement 1.** Structural comparison of MBLAC1 with ribonucleases from the MBL superfamily and with glyoxalase II.
DOI: https://doi.org/10.7554/eLife.39865.003

MBLAC1 and CPSF73 reveals similar di-zinc ion binding modes, there are clear differences in their active site flanking loops; only the β3-β4 MBLAC1 active site loop (β1-β2 in LACTB2), is present in CPSF73 (β1-β2 loop), where it is relatively shorter (*Figure 1—figure supplement 1D*).

## Loss of MBLAC1 impairs cell cycle progression

We sought to gain insight into the biological role of MBLAC1 by its depletion in HeLa cells using siRNA. Flow cytometry reveals that unsynchronised cells depleted for MBLAC1 show a cell cycle defect (*Figure 2A*). This is manifested by increased accumulation of cells in G$_1$/early S-phase and decreased proportions of cells in G$_2$ compared to controls. Furthermore, cells synchronised in early S-phase using a double thymidine block (*Harper, 2005*) and harvested immediately after block release show the expected G$_1$/early S phenotype (*Figure 2A*). However, 4 hours post-release, control cells progressed normally through the cell cycle, while MBLAC1 depleted cells displayed a strong delay in G$_1$/early S-phase. Western blots were then used to investigate the nature of the delay in cell cycle progression by monitoring levels of cyclin D1, a cell cycle protein marker expressed during G$_1$ (*Darzynkiewicz et al., 1996*) (*Figure 2B*). Cyclin D1 was hardly detectable in controls, but was strongly upregulated with MBLAC1 depletion (*Figure 2B*). These results are consistent with a relatively stronger G$_1$ block with MBLAC1 depletion compared to control cells (*Figure 2A–B*). Given a possible role for MBLAC1 in cell cycle progression, both unsynchronised and early S-phase synchronised wild-type HeLa cells (using a double thymidine block) were used to investigate the endogenous subcellular distribution of MBLAC1 during S-phase. Western blot analyses reveal the presence of MBLAC1 in both the cytosol and nucleus (*Figure 2C*). However, a fraction of MBLAC1 was consistently observed to localise in the nuclear compartment during early S-phase of the cell cycle (and less consistently at later S-phase time points), suggesting a cell-cycle-dependent nuclear function for MBLAC1.

Following the observation of nuclear localised MBLAC1 during S-phase and our crystallographic analyses which suggested MBLAC1 as a possible ribonuclease distinct from, but related to, CPSF73, both proteins were depleted in HeLa cells using siRNA to investigate the roles of MBLAC1 in S-phase. Incorporation of 5-bromo-2'-deoxyuridine (BrdU) into newly synthesised DNA in synchronised cells was monitored by flow cytometry (*Figure 3A*, left graph) followed by propidium iodide (PI) staining (*Figure 3—figure supplement 1*). The abundance of BrdU positive cells, corresponding to S-phase cells, was strongly reduced in MBLAC1 and CPSF73 knockdowns at early time points (30 min, 1 hr), compared to controls (*Figure 3A*, left graph). Subsequent time points displayed a reduced difference, though 6 hr post release a consistent pool of G$_1$ cells was still apparent in

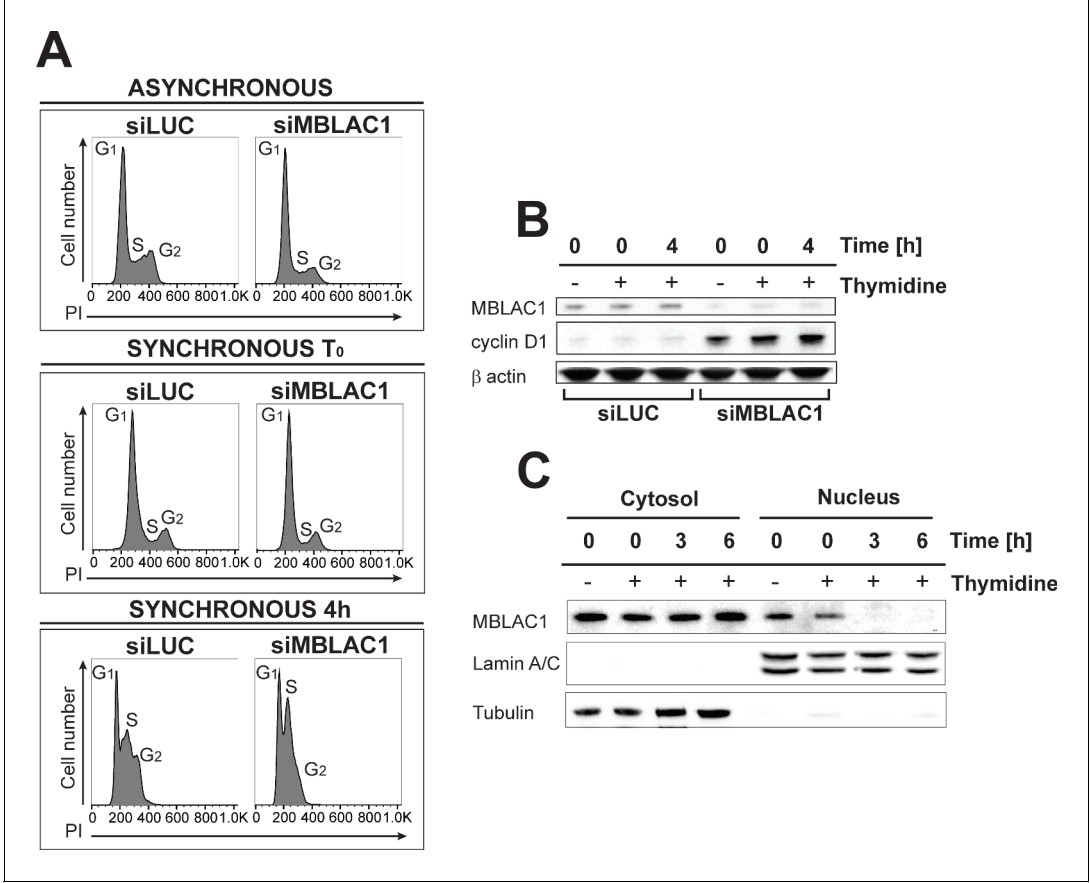

**Figure 2.** Loss of MBLAC1 impairs cell cycle progression. (**A**) Cell cycle analyses of MBLAC1-depleted cells. Analyses were on unsynchronised cells, or after cell synchronization (using a double thymidine block) as indicated. Flow cytometry profiles were obtained by PI staining. Control siRNA (siLUC) transfected cells were used for reference. (**B**) Western blots evaluating knockdown efficiency of MBLAC1 after siRNA treatment, in unsynchronised and synchronised cells. Cyclin D1 ($G_1$ marker) levels were analysed. β-Actin was used as a control. Control siRNA (siLUC) transfected cells were used for reference. (**C**) Western blot analysis showing cellular distribution of endogenous MBLAC1 after subcellular fractionation in unsynchronised (-) and early S-phase synchronised HeLa cells (+) at different time points after release from a double thymidine block. Note the presence of MBLAC1 in the nucleus in early S-phase. α-Tubulin and lamin A/C were used as cytosolic and nuclear markers, respectively.

DOI: https://doi.org/10.7554/eLife.39865.005

MBLAC1-depleted cells compared to both control and CPSF73-depleted cells (*Figure 3—figure supplement 1*). These results indicate that both MBLAC1 and, as expected, CPSF73 depleted cells are impaired in efficiently entering/progressing in S-phase. Although both depletions displayed a delay 30 min post-release, the defect was more pronounced with MBLAC1 depletion over the full cell cycle, implying a cell-cycle-specific role for MBLAC1 (*Figure 3—figure supplement 1*). Western blot analyses indicated that in control samples (siLuc) MBLAC1 levels were increased 6 hr post-release (*Figure 3B*) corresponding to mid S-phase (as shown by flow cytometry profiles, *Figure 3—figure supplement 1*). These observations support a specific role for MBLAC1 during S-phase. By contrast, CPSF73 expression levels were relatively constant across the cell cycle (18 hr), consistent with its broad role in pre-mRNA processing (*Mandel et al., 2006*) (*Figure 3B*). Although both MBLAC1 and CPSF73-depleted cells progress through the complete cell cycle, levels of cyclins D1 ($G_1$ marker) and E (a $G_1$/S boundary and S progression marker) were strongly modified compared to controls. As described above, cyclin D1 levels were strongly upregulated with MBLAC1 depletion, while cyclin E levels were reduced compared to controls. Notably for CPSF73 depletion, cyclins D1 and E showed intermediate levels compared to MBLAC1 depletion and controls (*Figure 3B*). Given the strong correlation between S-phase and histone biosynthesis (*Harris et al., 1991*; *Ewen, 2000*), histone H3 levels were used to analyse RD histone protein abundance in MBLAC1 and CPSF73 depleted cells. Notably, H3 levels were reduced in MBLAC1 depleted cells; by contrast, CPSF73

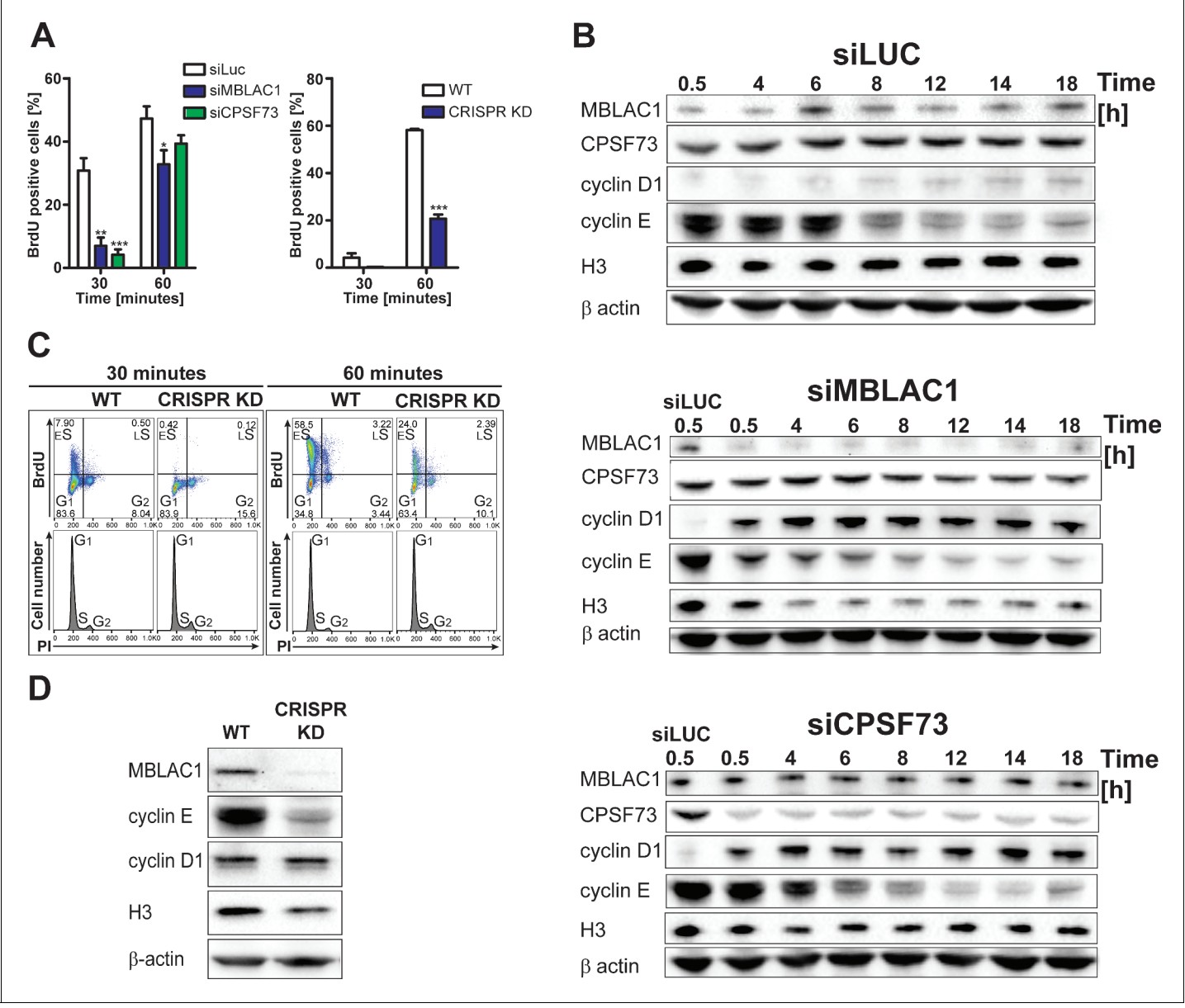

**Figure 3.** Loss of MBLAC1 impairs normal entrance to S-phase and S-phase progression. (**A**) Comparative quantification of BrdU-positive cells corresponding to early-S cells in MBLAC1 and CPSF73-depleted cells and MBLAC1 CRISPR KD (CRISPR/Cas9 mediated KD) cells after synchronization. Control siRNA (siLUC) transfected cells and wt HeLa cells were used for comparison. Error bars represent SEM from three independent biological replicates. Asterisks indicate the statistical significance using the two way ANOVA tool followed by the Bonferroni multiple comparison tool in Graphpad Prism five software: *, $p<0.05$; **, $p<0.001$; ***, $p<0.0001$. Unlabeled variations were considered not statistically relevant. (**B**) Western blot analyses evaluating MBLAC1 and CPSF73 knockdown efficiency and effects on cyclin D1, cyclin E and histone H3 levels. (**C**) Cell cycle analysis of MBLAC1 CRISPR/Cas9 mediated stable knockdown (CRISPR KD) cells after synchronization in early S-phase. Flow cytometry profiles obtained by double-staining with an anti-BrdU antibody (upper panels) and PI (lower panels). wt HeLa cells were used for reference. Abbreviations: $_ES$, early S-phase; $_LS$, late S-phase. (**D**) Western blot analyses evaluating MBLAC1 CRISPR/Cas9-mediated stable knockdown efficiency and effects on cyclin D1, E and histone H3 levels (with synchronised cells treated with BrdU, 30-min incubation).

DOI: https://doi.org/10.7554/eLife.39865.006

The following figure supplements are available for figure 3:

**Figure supplement 1.** Loss of MBLAC1 impairs normal entering of S-phase and S-phase progression.
DOI: https://doi.org/10.7554/eLife.39865.007

**Figure supplement 2.** MBLAC1 CRISPR/Cas9 stable knockdown and FLAG knock in validation, wild-type-like cell phenotype restoration and MBLAC1 interactors.
DOI: https://doi.org/10.7554/eLife.39865.008

depletion showed an intermediate H3 level compared to controls (*Figure 3B*). A more severe pheno-type was observed in an MBLAC1 stable knockdown (*Figure 3A right graph, C, D*) (*Figure 3—figure supplement 2A–C*) generated in HeLa cells by CRISPR/Cas9 gene editing. This observation supports a specific role for MBLAC1 in cell cycle regulation, in particular on entering/progressing through S-phase (*Figure 3A right graph, C*). As for the siRNA-mediated knockdown experiments, CRISPR/Cas9 depletion of MBLAC1 results in an altered production of cyclin E and reduced level of H3. By contrast, cyclin D1 levels appear unchanged, possibly due to a mechanism of cellular adaptation under unfavourable conditions (*Figure 3D*).

Treatment of MBLAC1 CRISPR/Cas9-mediated knockdown (CRISPR KD) cells with isolated recombinant MBLAC1 partially restored the wild-type cell phenotype leading to dose-dependent increasing levels of cyclin E and histone H3 compared to untreated MBLAC1 KD cells after synchronization in early S-phase (*Figure 3—figure supplement 2D*). These observations support a direct role for MBLAC1 during S-phase of the cell cycle. The subcellular distribution of ectopically administered recombinant MBLAC1 in CRISPR/Cas9-mediated knockdown cells appears to be very similar compared to that of wild-type HeLa cells (*Figure 3—figure supplement 2E*), indicating that application of recombinant MBLAC1 partially restores the wild-type like phenotype by localizing to the correct cellular compartment during S-phase.

## MBLAC1 is involved in histone-specific pre-mRNA processing in vivo

Given the established role of CPSF73 as a nuclear ribonuclease responsible for pre-mRNA 3' end maturation (*Dominski et al., 2005*; *Mandel et al., 2006*), we investigated such a role for MBLAC1. MBLAC1 was depleted in HeLa cells using siRNA as, for comparison, was CPSF73 (*Figure 4A*). To evaluate not only the polyadenylated mRNA population (polyA), but also RD histone mRNA (SL) processing, cells were synchronised in early S-phase and chromatin associated RNA was extracted to capture nascent transcripts (*Nojima et al., 2015*). Chromatin RNA-seq (ChrRNA-seq) analyses for a typical polyA mRNA encoding gene, for example glyceraldehyde-3-phosphate dehydrogenase (GAPDH), displayed clear transcription termination defects downstream of the transcription end site (TES) in CPSF73, but not MBLAC1, depleted cells, nor in a siRNA controls (*Figure 4B*; *Figure 4—figure supplement 1A*). Unexpectedly, RD histone genes (SL) showed apparent transcription termination defects, not only in CPSF73, but also in MBLAC1, depletions (*Figure 4C*; *Figure 4—figure supplement 1B*). Importantly, 43 non-overlapping RD histone genes were analysed; of these 29 RD histone genes manifested clear transcription termination defects, in both the MBLAC1 and CPSF73-depleted samples (evidence for/exceptions are HIST1H2AI and HIST4H4 genes on which CPSF73 depletion apparently does not show an effect); a possible effect was observed on 7 RD histone genes (*Figure 4—figure supplements 2,3*), and no apparent effect was detected on seven remaining RD histone genes (*Figure 4—figure supplements 2,4*). Where observed, read through (RT) transcripts corresponding to this defect were detectable covering ~200 bases downstream of the transcription end site (TES) with the RD histone genes (*Figure 4—figure supplements 2,3*). The comparison of the read through of MBLAC1 to that of CPSF73 depletion shows a similar transcription termination defect pattern affecting similar genes with a few exceptions (*Figure 4—figure supplements 2,4*); it is possible that both MBLAC1 and CPSF73 may affect overlapping sets of histone genes, but to different extents.

Quantitative reverse transcription PCR (RT-qPCR) analyses validated the transcriptomics results. With normal protein-coding mRNA (GAPDH), transcription termination was substantially affected by CPSF73, but not MBLAC1, depletion (*Figure 4D*). The abundance of termination defective (3' end extended) RD histone mRNA increased in both MBLAC1- and CPSF73-depleted cells compared to controls (*Figure 4D*) supporting the proposal that MBLAC1 is a RD histone selective pre-mRNA processing enzyme. Consistent with this proposal, the relative levels of defective transcripts quantified by RT-qPCR analyses reflected the difference observed during ChrRNA-seq analyses between different replication-dependent histone genes (e.g. a stronger defect was observed on HIST1H2BC, compared to HIST4H4, mRNA) (*Figure 4C,E*; *Figure 4—figure supplement 2B*).

Single gene analyses on chrRNA confirmed the increased abundance of termination defective (3' end extended) RD histone mRNA in MBLAC1 CRISPR/Cas9-mediated knockdown cells compared to wildtype HeLa cells, after synchronization in early S-phase (*Figure 4E*). This result supports the observed altered cell cycle phenotype as observed following depletion of MBLAC1 using siRNA (*Figures 2* and *3*). To investigate the nature of these defective transcripts, total RNA extracted from

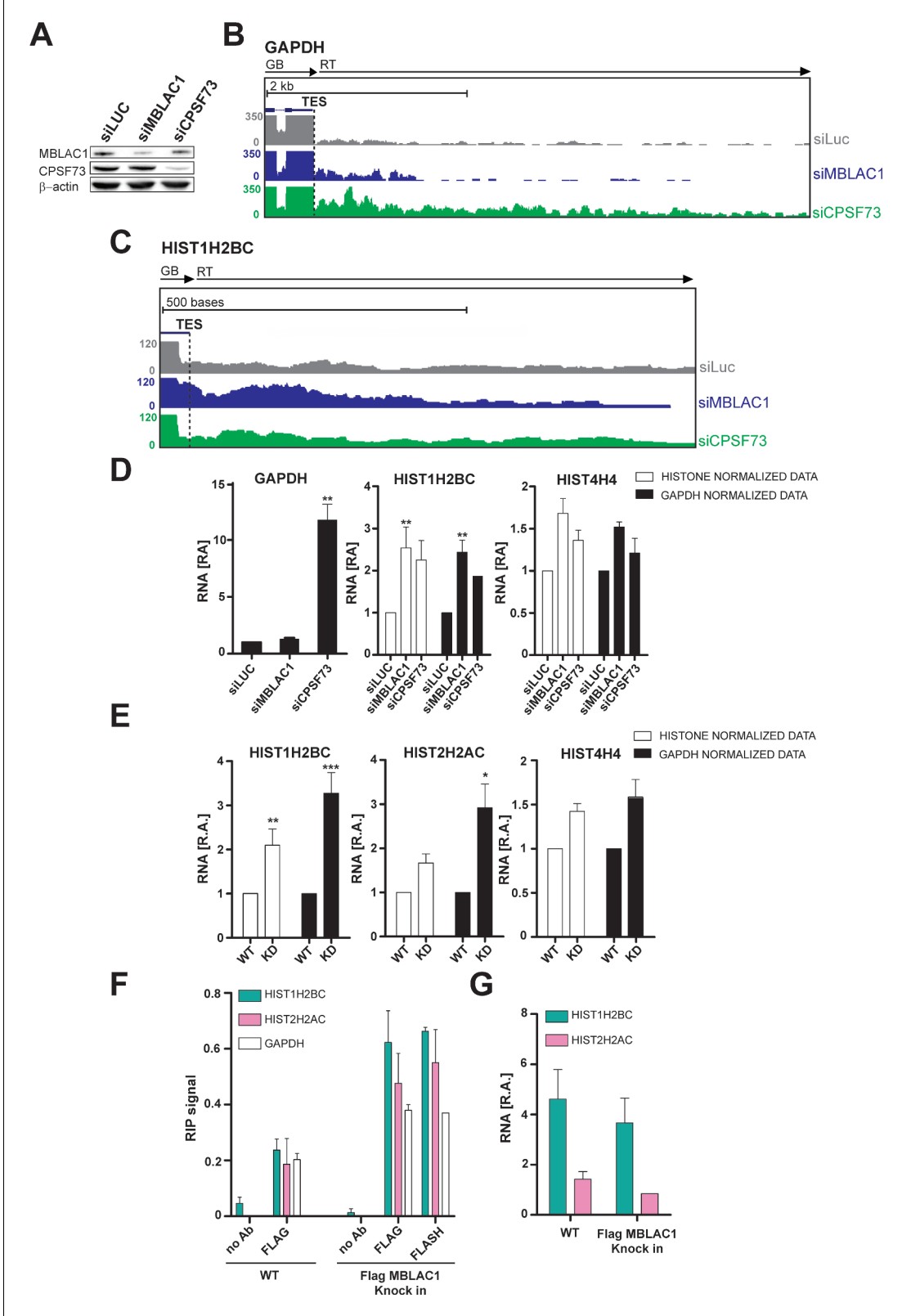

**Figure 4.** Loss of MBLAC1 leads to 3' end processing defects in replication dependent histone pre-mRNA. (**A**) Western blots showing the knockdown efficiency of siRNA treatments for MBLAC1 and CPSF73 in synchronised HeLa cells (using β-actin as a control). (**B, C**) ChrRNA-seq analyses from one of two biological replicates (Set2), on GAPDH or selected RD histone (HIST1H2BC) genes (UCSC Genome Browser) (**Kent et al., 2002**), following MBLAC1 or CPSF73 depletion. Arrows indicate gene bodies and transcription termination defect orientation. (**D**) Real-time qPCR quantification of 3' end

*Figure 4 continued on next page*

*Figure 4 continued*

transcription termination defect following MBLAC1 and CPSF73 depletion on genes as in (**b, c**). Relative abundance (RA) of unprocessed RNA was normalised to the abundance of GAPDH mRNA (black), or to that of the corresponding histone gene body (white) (*Table 2*). Error bars represent SEM from four biological replicates. Asterisks indicate the statistical significance observed using the two-tailed *t*-test (\*\*p<0.01). Abbreviations: GB, gene body; RT, read through; TES, transcription end site. (**E**) Real Time-qPCR quantification of 3' end transcription termination defect in the MBLAC1 CRISPR/Cas9-mediated stable knockdown (KD) compared to wt HeLa cells. The relative abundance (RA) of unprocessed RNA was normalised to GAPDH mRNA (black), or to the corresponding histone gene body abundance (white) (*Table 2*). Error bars represent SEM from four biological replicates. Asterisks indicate the statistical significance observed using the two-tailed *t*-test (\*,p<0.05, \*\*, p<0.01, \*\*\*, p<0.001). Unlabeled variations were considered likely not statistically relevant. (**F**) RNA-Chromatin immunoprecipitation (RNA-ChIP) analyses investigating the recruitment of MBLAC1 on histone RNA. RNA-ChIP signals are shown for MBLAC1 and FLASH in the MBLAC1 C-terminal FLAG knock in HeLa cells; the FLAG background signal is shown in wild-type cells used as control for comparison. Note that both MBLAC1 and FLASH signals are comparable on normal protein coding RNA (GAPDH). Error bars represent SEM from three (MBLAC1) or two (FLASH) biological replicates. (**G**) RNA-ChIP input samples relative comparison. Histone mRNA relative abundance (in both wild-type and MBLAC1 C-terminal FLAG knock HeLa cells) was normalised to the abundance of GAPDH mRNA in the corresponding sample. The observed results imply that the RD histone mRNA levels are not affected (within detection limits) by the tagging of MBLAC1. Error bars represent SEM from three biological replicates.

DOI: https://doi.org/10.7554/eLife.39865.009

The following figure supplements are available for figure 4:

**Figure supplement 1.** 3' end processing defects observed in replication-dependent histone pre-mRNA after MBLAC1 depletion and investigation of the effect of MBLAC1 depletion on the polyadenylated fraction of histone mRNA in HeLa cells.

DOI: https://doi.org/10.7554/eLife.39865.010

**Figure supplement 2.** MBLAC1 and CPSF73 depletions affect RD histone pre-mRNA processing in vivo.

DOI: https://doi.org/10.7554/eLife.39865.011

**Figure supplement 3.** MBLAC1 and CPSF73 depletions affect RD histone pre-mRNA processing in vivo.

DOI: https://doi.org/10.7554/eLife.39865.012

**Figure supplement 4.** MBLAC1 and CPSF73 depletions affect RD histone pre-mRNA processing in vivo.

DOI: https://doi.org/10.7554/eLife.39865.013

early S-phase synchronised HeLa cells (wildtype and the MBLAC1 CRISPR/Cas9-mediated knock-down) was used to evaluate the accumulation of defective histone pre-mRNAs in either the polyade-nylated (polyA+) or unpolyadenylated (polyA-) RNA cellular fractions (*Figure 4—figure supplement 1C–E*). Quantitative reverse transcription PCR (RT-qPCR) analyses showed a low (~15%), but consistent, increase in polyA +histone transcripts in MBLAC1-depleted cells compared to wild-type cells (*Figure 4—figure supplement 1C*). By contrast, in both wild-type and MBLAC1-depleted cells, the polyA- fraction showed a similar, or reduced, abundance of histone transcripts. These observations support previous studies (*Sullivan et al., 2009b*; *Romeo et al., 2014*; *Kari et al., 2013*), by implying that RD histone pre-mRNA can undergo polyadenylation when inefficiently processed at their normal 3' processing site.

To further investigate the involvement of MBLAC1 in RD histone pre-mRNA cleavage in vivo, RNA-Chromatin immunoprecipitation (RNA-ChIP) experiments (*Sun et al., 2006*) (*Figure 4F–G*) were carried out using a C-terminal FLAG tagged MBLAC1 knock in generated in HeLa cells by CRISPR/Cas9 technology (*Figure 3—figure supplement 2F, G*). The results provided evidence that MBLAC1 specifically associates with chromatin associated histone mRNA during early S-phase, but that it either does not interact, or interacts to a lesser extent (similarly to FLASH) (*Figure 4F–G*), with normal protein coding mRNA (GAPDH); a similar amount of FLASH was recruited on histone RNA compared to MBLAC1. FLASH was used as control in our experiments as it is a well-established component of the histone 3' end processing complex (*Marzluff and Koreski, 2017*) (*Figure 4F*). Note that RD histone mRNA relative levels are not affected by the tagging of MBLAC1 as observed by comparison of wild-type and MBLAC1 C-terminal FLAG knock HeLa cells RNA-ChIP input samples (*Figure 4G*).

## MBLAC1 shows endoribonucleolytic activity in vitro

The combined observations described above indicated that MBLAC1 is a ribonuclease, which is possibly involved in RD histone pre-mRNA processing. We therefore purified recombinant wild-type MBLAC1 (wtMBLAC1) and a H196A/D221A/H263A variant (mutMBLAC1) (with disrupted zinc ion binding) (*Figure 5A–B*) from *E. coli*, as well as wtMBLAC1 produced in HEK293 cells. These protein preparations were tested for ribonuclease activity using an internally $\alpha$ $^{32}$P UTP labeled RNA

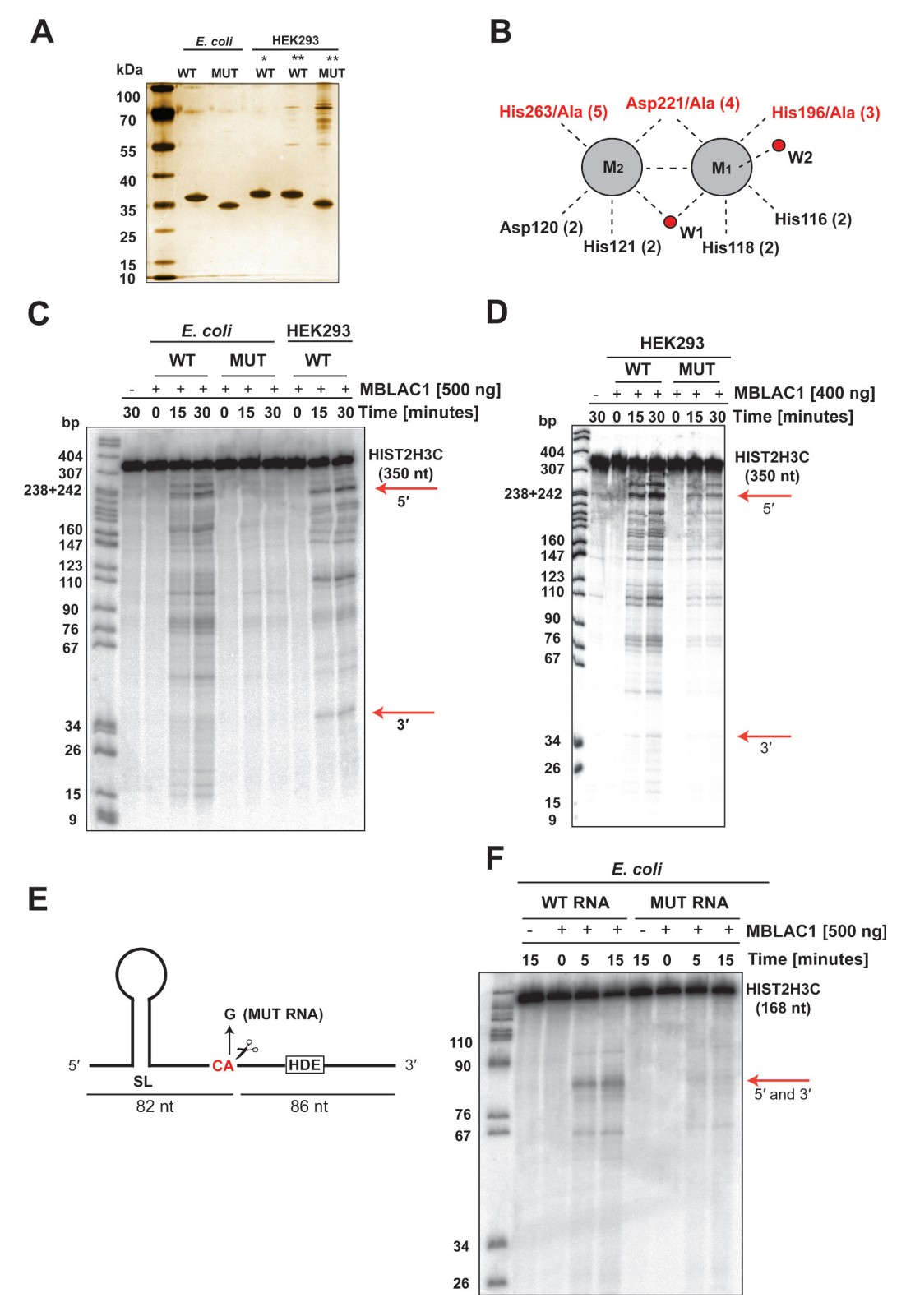

**Figure 5.** MBLAC1 has sequence-specific endoribonucleolytic activity on RD histone pre-mRNA. (**A**) Silver staining analyses showing purity of MBLAC1 protein preparations produced from different sources. Note all proteins are highly purified except mutMBLAC1 produced in HEK293 which presents some contaminants. (**B**) Schematic view of metal binding and active site residues and the two water molecules (red spheres) observed to coordinate (black dashed lines) to the metals (grey spheres) at the MBLAC1 active site by crystallography (numbering as in *Figure 1*). Active site residues

*Figure 5 continued on next page*

*Figure 5 continued*

substituted to alanine in the MBLAC1 variant (H196A/D221A/H263A) are in red. (C) Time-dependent in vitro cleavage assay using an internally labeled [32P] histone 2H3C pre-mRNA fragment (350 nt) with *E. coli* produced wtMBLAC1, or an active site substituted MBLAC1 variant (H196A/D221A/H263A), or HEK293 produced wtMBLAC1 (media secreted). Note only a fraction of the total RNA is hydrolyzed. (D) Time-dependent in vitro cleavage assay using an internally labeled [32P] histone 2H3C pre-mRNA fragment (350 nt) with HEK293 produced wtMBLAC1, or an active site substituted MBLAC1 variant (H196A/D221A/H263A). The red arrows indicate the assigned position of 3' end processing in vivo. Note that both wt and mutMBLAC1 used in this experiment were purified from HEK293 cell lysates instead of from cell media due to the loss of secretion of the mutant enzyme in the cell media. (E) Schematic view of the histone 2H3C pre-mRNA fragment (168 nt) used in the cleavage assays showing preferential cleavage occurs after the CA dinucleotide located five nucleotides to the 3' side of the stem loop. The single-nucleotide substitution (A/G) at the CA cleavage site is shown. (F) Time dependence of *E. coli* produced wtMBLAC1 in vitro cleavage using an internally labeled [32P] unmodified (WT RNA), or with a single-nucleotide substituted (MUT RNA) histone pre-mRNA fragment (as in E). The red arrows indicate the assigned position of 3' end processing in vivo. Abbreviations: HDE, histone downstream element; SL, stem loop.

DOI: https://doi.org/10.7554/eLife.39865.014

substrate corresponding to a 350 nt 3' positioned fragment of the human HIST2H3C gene, including its 3' processing elements (*Figure 5C–D*). Recombinant wtMBLAC1 purified from *E. coli* bacteria displayed significant nonspecific ribonucleolytic activity, degrading the substrate in a time dependent manner. By contrast, mutMBLAC1 manifested very little activity over time under the same assay conditions. Notably, the endoribonucleolytic activity of wtMBLAC1 purified from HEK293 cells using the same $\alpha$ 32P UTP-labeled histone RNA fragment gave an apparently more specific cleavage pattern. In particular, cleavage products corresponding in size to the biologically authentic 5' and 3' fragments were manifested (*Figure 5C*) (shown by red arrows). This apparently more specific cleavage by wtMBLAC1 when produced in HEK293 cells may reflect different kinetic properties, possibly resulting from post-translational modifications, or more biologically correct metal complexation at the active site (*Figure 1C*). Note that bacterial MBLfold hydrolases are active with Zn (II) and other transition metals including Fe (II) (*Cahill et al., 2016*). To further exclude the possibility of contaminating nuclease activity in our protein preparations and therefore validate MBLAC1 endoribonucleolytic activity, the previously described RNA substrate was also incubated with wtMBLAC1 and the H196A/D221A/H263A variant (mutMBLAC1) purified from HEK293 cells (*Figure 5D*). Consistent with the previous results using bacterial produced enzymes, we observed little residual activity when the substrate was incubated with mutMBLAC1 compared to wtMBLAC1. We again observed a more specific cleavage pattern with wtMBLAC1 from HEK293 cells rather than with the bacterial produced wtMBLAC1 (*Figure 5D*).

We next investigated the potential cleavage specificity of MBLAC1 using a shorter $\alpha$ 32P UTP internally labeled RNA substrate (WT RNA) and a variant with a single-nucleotide substitution (A/G) at the major physiological histone pre-mRNA processing site (MUT RNA) (*Figure 5E*) (*Furger et al., 1998*; *Kolev and Steitz, 2005*). Bacterially produced wtMBLAC1 showed clear ribonucleolytic activity with the WT RNA substrate in a time-dependent manner (*Figure 5F*). Major cleavage products were observed (red arrow) at positions corresponding to the authentic 3' processing position. However, little evidence for ribonucleolytic activity was observed when wtMBLAC1 was incubated with MUT RNA (*Figure 5F*). This observation suggests that MBLAC1 specifically recognises the biologically authentic CA dinucleotide required for histone pre-mRNA maturation and that this site is critical for recognition/initial cleavage/binding by MBLAC1. Mononucleotides were not observed to accumulate under any of the tested conditions (data not shown) implying a lack of exonucleolytic activity for MBLAC1. Thus, consistent with the crystallographically observed structural similarity between MBLAC1 and LACTB2 (*Levy et al., 2016*) (*Figure 1H*), these results identify MBLAC1 as a ribonuclease with endonucleolytic activity. Unlike purified CPSF73 (*Mandel et al., 2006*), which exhibits substantially sequence-independent ribonuclease activity, isolated MBLAC1 appears more sequence selective especially with shorter RNA substrates (*Figure 5F*). These observations imply that, at least in isolated form, MBLAC1 initially catalyses hydrolysis at the biologically validated CA cleavage site with subsequent endonucleolytic cleavage occurring at other positions.

## RNA elements required for MBLAC1 specific activity in vitro

Specific elements are required for maturation of histone pre-mRNA in vivo (*Marzluff and Koreski, 2017*); in particular, SLBP appears to be necessary for efficient processing through stabilization of

the HDE-U7 snRNA interaction and the HLF complex involving symplekin, CPSF73, CPSF100, and CstF64, which are required for the cleavage of the pre-mRNA (*Dominski et al., 1999*; *Kolev et al., 2008*; *Romeo et al., 2014*). To investigate the specificity of the observed MBLAC1 hydrolytic activity on histone pre-mRNA, 50 to 40 nucleotide RNA fragments corresponding to a portion of the RD human histone HIST2H3C pre-mRNA were [γ-³²P] ATP 5' labeled and incubated with wtMBLAC1 produced in HEK239 cells (50 ng) (*Figure 6*). The reaction catalysed by MBLAC1 on the wild-type RNA 1 generated two main products assigned as 27 and 29 nucleotides long (based on our fragment numbering system), corresponding to the 'physiological' major and minor cleavage sites (*Streit et al., 1993*; *Furger et al., 1998*), respectively (*Figure 6A*). This observation is in agreement with previous studies, where the major and minor cleavage sites were reported to occur 5 and 7 nucleotides after the stem-loop, respectively, that is ACCCA corresponding to cleavage in position 27, and ACCCACA corresponding to cleavage in position 29 (*Scharl and Steitz, 1994*; *Furger et al., 1998*). When a single nucleotide in RNA 1 is mutated at position 27 (A/G), to abolish the 'physiological' CA dinucleotide at this position (RNA 2), we observed production of a single main processing product corresponding to cleavage at position 29 (*Figure 6A*). Although hydrolysis of RNA 2 was clearly observed, it appeared less efficient compared to that of RNA 1. This observation is in agreement with the results described above where a single point mutation at position 27 negatively affects catalysis (*Figure 5F*). mutMBLAC1 purified from HEK293 cells was also tested using RNA 2 to investigate its ability to hydrolase the substrates at specific positions; as anticipated, mutMBLAC1 showed very little hydrolytic activity on RNA 2 compared to wtMBLAC1 (*Figure 6B*). To further investigate the specificity of cleavage after the CA at position 27, the wild-type RNA was modified to contain only one CA dinucleotide at position 27 (RNA 3). Incubation with MBLAC1 led to the production of a single observed degradation product consistent with cleavage at position 27 (major cleavage site), but again with apparently impaired efficiency (*Figure 6C*). When the nucleotide sequence of the stem-loop was inverted, to maintain secondary structure (RNA 4), the cleavage reaction appeared less efficient, although the cleavage specificity was apparently not compromised within our limits of detection (note that the processing products of RNA 4 has a slower electrophoretic mobility) (*Figure 6D*). This observation suggests the precise stem-loop sequence composition does not affect the correct positioning of MBLAC1 at the physiological CA cleavage site. When the stem-loop sequence was modified to ablate its conformation (RNA 5), cleavage at position 27 (the major cleavage site) was apparently completely abolished; instead, a major band was observed corresponding to cleavage at the CA, position 23, although longer degradation fragments were also detected (consistent with cleavage at positions 29, and possibly 32, 35) (*Figure 6E*). The degradation patterns observed for RNA 4 and 5 imply that likely the stem-loop structure prevents MBLAC1 from accessing putative single-stranded cleavage sites located upstream of position 27, thus helping to maintain specificity of cleavage at position 27. MBLAC1 activity was also evaluated on a fragment lacking the HDE sequence (RNA 6); interestingly, with this substrate efficient cleavage was maintained and the major cleavage site (position 27) still recognised, although the cleavage specificity was reduced, leading to the formation of additional degradation bands corresponding to hydrolysis at all the CA dinucleotides present in the fragment 6 sequence (*Figure 6F*). Overall, these results support the proposal that purified MBLAC1 displays significant histone mRNA 3' end processing specificity.

## Discussion

The combined biochemical, cellular, and genetic analyses presented here provide clear evidence that MBLAC1 is an endoribonuclease selective for 3' end processing of RD histone pre-mRNA during S-phase of the cell cycle. Assays with isolated recombinant MBLAC1 reveal it is selective for hydrolytic cleavage at the biologically relevant CA dinucleotides present 5 and 7 nucleotides downstream of the stem-loop sequence in replication-dependent pre-mRNA (positions 27 and 29, respectively, using our pre-mRNA numbering). Crystallographic and other biophysical analyses support the assignment of MBLAC1 as a nuclease, which is related to CPSF73 in terms of the overall fold of its MBL domain, but which has distinctive, structural features, possibly reflecting their different substrate recognition and/or substrate binding modes (*Figure 1* and *Figure 1—figure supplement 1*).

The results obtained by depletion of MBLAC1 using siRNA or CRISPR/Cas9 techniques support the involvement of MBLAC1 in RD histone pre-mRNA processing in vivo. The slow growth rate of

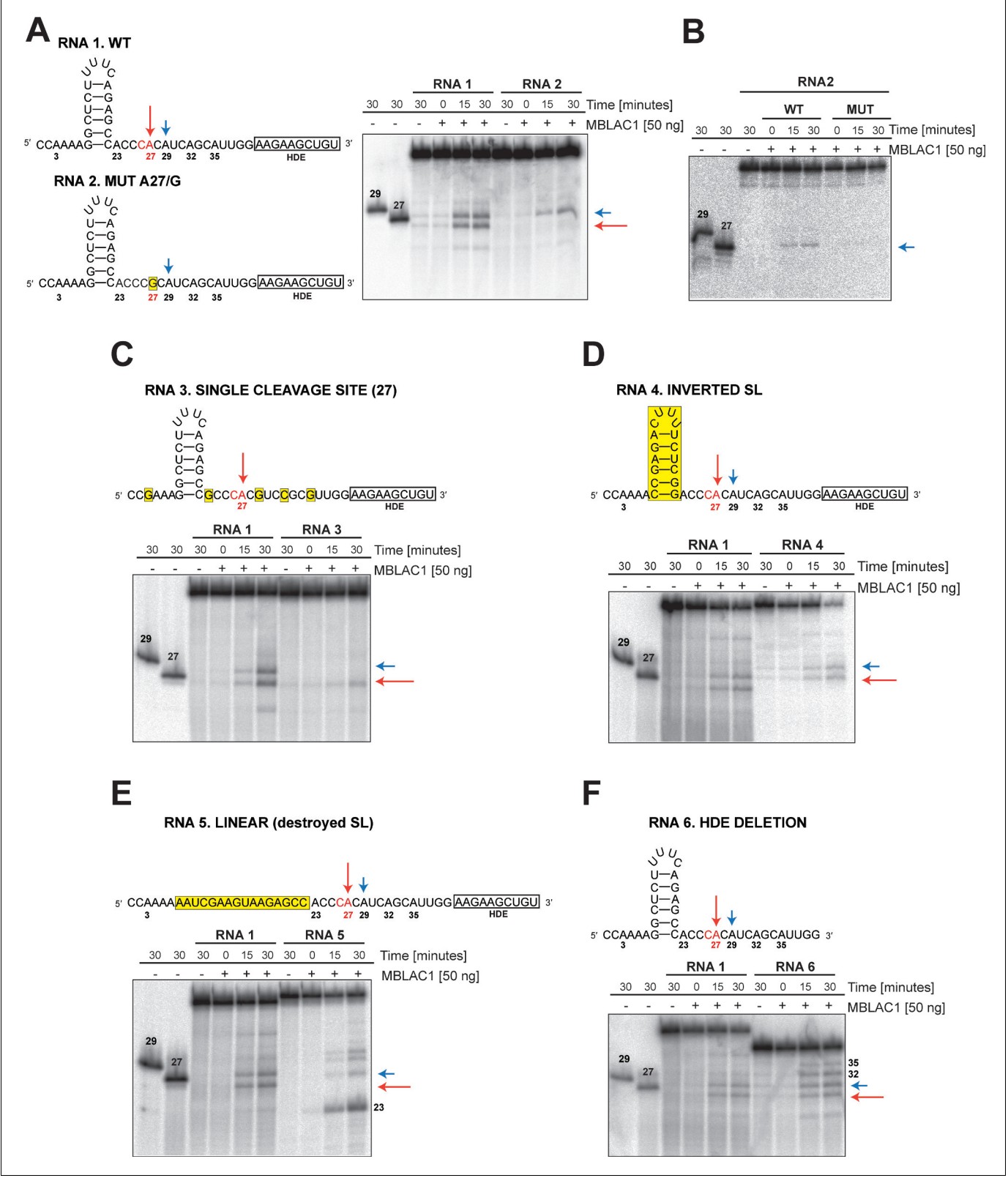

**Figure 6.** MBLAC1 shows specific endoribonucleolytic activity on histone pre-mRNA recognising specific sequence/structural RNA elements in vitro. (A, C–F) Time-dependent in vitro cleavage assays using [γ-$^{32}$P] ATP 5' labeled histone 2H3C RNA fragments (RNA 1–6) in the presence of HEK293 produced wtMBLAC1. (B) Time-dependent in vitro cleavage assays using [γ-$^{32}$P] ATP 5' labeled histone 2H3C RNA 2 (50 nt) in the presence of HEK293 produced wild-type or mutMBLAC1. Cleavage products of RNA 2 after incubation with wtMBLAC1 (WT) are compared with those after incubation with

*Figure 6 continued on next page*

*Figure 6 continued*

mutMBLAC1 (MUT). Cleavage products of the RNA 1 are compared with those of the differently modified fragments. Schematic views of the RNA fragments are shown. The schematics show the numbering assigned to the possible cleavage sites present in each RNA; arrows indicate the physiologically major (red) and minor (blue) cleavage sites. The CA dinucleotide corresponding to the major cleavage site is in red. Sequence or structural modifications of each RNA are in yellow. The size of processed RNA products is indicated when appropriate. Unprocessed RNA fragments corresponding to nucleotides 1 to 29 and 1 to 27 of the WT RNA 1 were used as mass markers and indicated as 29 and 27, respectively. Note that the 29 and 27 markers migrate more slowly on gel electrophoresis compared to the corresponding fragments generated by MBLAC1 catalysis in positions 27 and 29. This observation is explained by the presence of an extra negatively charged phosphate group on the 5'end of the products generated by MBLAC1 catalysis, compared to the markers, as expected during maturation of histone transcripts (*Furger et al., 1998*).

DOI: https://doi.org/10.7554/eLife.39865.015

MBLAC1-depleted cells (*Figure 3—figure supplement 2*) likely reflects its important role in RD histone pre-mRNA processing which is required for normal progression through the cell cycle. The observation of a cell cycle defect in entering/progressing through S-phase correlates with reduced histone protein abundance in HeLa cells after depletion of MBLAC1, either using siRNA or CRISPR/Cas9 (*Figure 3*). These results are in agreement with a defect in efficient histone pre-mRNA maturation as we observed by chrRNA-seq analyses in HeLa cells depleted of MBLAC1 where RD histone pre-mRNA transcripts manifested abnormally 3' end extensions reflecting inefficient maturation. The inability of MBLAC1-depleted cells to produce the appropriate amount of histone proteins due to the lack of mature mRNA encoding for them is also consistent with occurrence of a 'G$_1$-like' block, as observed by flow-cytometry and western blot analyses (*Figure 3* and *Figure 3—figure supplement 1*). These results imply that during S-phase, MBLAC1 has a specialised role in RD histone pre-mRNA processing.

Studies with recombinant MBLAC1 purified from human or *E. coli* cells, define MBLAC1 as an endoribonuclease, with selectivity for cleavage at the biologically authentic CA dinucleotides (positions 27 and 29 using our nomenclature, *Figure 6*). These results imply that the stem-loop secondary structure of RD nascent pre-mRNA is involved in enabling preferential cleavage at position 27. The presence of the HDE sequence appears to be required to avoid non-specific cleavage, although cleavage at position 27 was still observed when the HDE was removed (*Figure 6F*). The involvement of the HDE element in determining selectivity is supported by in vitro studies utilizing HeLa nuclear extracts indicating that the HDE is a major factor in defining the cleavage site of RD histone pre-mRNA (*Scharl and Steitz, 1994*). The work with isolated MBLAC1 reveals that it has potential to catalyse cleavage at CA sites other than at positions 27 and 29, although the biological relevance of these other reactions needs to be validated. Future work can also focus on establishing the details of the kinetics of MBLAC1 catalysis and the structural features that enable its selectivity, including the nature of its interactions with the stem-loop and the HDE element at the 3' end of RD histone pre-mRNA.

Further detailed work is required to compressively define the factors responsible for targeting MBLAC1 to the RD histone pre-mRNA cleavage site. The available results indicate that it is possible that the observed in vitro sequence specificity of MBLAC1 may enable recognition of core histone pre-mRNA substrates without the requirement for as many additional factors as is necessary in the case of CPSF73 (or that MBLAC1 requires different factors) (*Sullivan et al., 2009b*).

Consistent with previous reports (*Dominski et al., 2005*; *Kolev et al., 2008*; *Sullivan et al., 2009a*), we found that CPSF73 promotes RD histone pre-mRNA processing. However, it is likely that CPSF73 is relatively more important than MBLAC1 in RD histone pre-mRNA processing outside of S-phase, reflecting its essential role in the biosynthesis of polyadenylated mRNA, including that of multiple histone variants (*Marzluff et al., 2008*; *Marzluff et al., 2002*; *Wagner et al., 2007*). CPSF73 could well also be wholly or partly responsible for processing of defective RD histone pre-mRNA that subsequently undergoes polyadenylation (*Sullivan et al., 2009b*) when MBLAC1 is depleted in cells. Reported biochemical work with CPSF73 in its isolated form has only been carried-out with few histone pre-mRNA model sequences (*Kolev and Steitz, 2005*). Thus, it is possible that both CPSF73 and MBLAC1 selectively impact on the rates of maturation of different histone pre-mRNA substrates in vivo, opening up the possibility for selective modulation of the biosynthesis of RD histone variants. Although further validation work is required, our observations in cells support

also the proposal that MBLAC1 and CPSF73 impact to different extents on similar genes, possibly compensating for depletion of one another in cells (*Figure 4—figure supplement 2*).

Our genetic and cellular studies imply that MBLAC1 inhibition may selectively slow down S-phase progression in cancer cells. Thus, MBLAC1 may represent a new S-phase specific cancer target. The true bacterial MBLs involved in antibacterial resistance have been shown to be amenable to small molecule targeting (*Pettinati et al., 2016*), and it is likely that human MBL-fold endonucleases will also be sensitive to small molecule inhibition. Our results suggest that targeting MBLAC1 may not be as toxic as CPSF73, which has a pleiotropic role in polyadenylated pre-mRNA processing (*Mandel et al., 2006*). Thus, the development of MBLAC1 inhibitors, likely for use in combination with other drugs, is of interest from the cancer treatment perspective. Finally, it is notable that relatively non-selective zinc ion chelating compounds, including HDAC inhibitors such as SAHA (suberanilohydroxamic acid) (*West and Johnstone, 2014*), are already used for cancer treatment via S-phase targeting. It is possible that these, at least in part, work by inhibiting metal dependent endoribonucleases, such as CPSF73 and MBLAC1.

## Materials and methods

### Reagents and antibodies

Unless otherwise specified, reagents were from *Sigma-Aldrich*. Antibodies used were: goat anti-MBLAC1 (sc-243427), HRP-conjugated donkey anti-goat (sc-2020), goat anti-Lamin B1 (sc-30264), mouse anti-Cyclin D (sc-20044), mouse anti-Cyclin E (sc-247) (all from *Santa Cruz Biotechnology, Inc.*); rabbit anti-MBLAC1 (ARP70842_P050) (*Aviva System Biology*); mouse anti-His tag (ab18184), rabbit anti-His tag (ab9108), mouse anti-Histone H3 (ab10799), HRP-conjugated mouse anti-β-actin (ab49900) (all from *Abcam*); HRP-conjugated goat anti- rabbit (170–6515) (*Biorad*); rabbit anti-CPSF73 (A301-091A) (*Bethyl Laboratories, Inc,*); mouse anti-Lamin A/C (4777) (*Cell Signaling Technology*); mouse anti-FLAG M2 (F3165) and mouse anti-α-tubulin (T5168) (*Sigma-Aldrich*); HRP-conjugated goat anti-mouse (W402B) (*Promega Corporation*); FITC mouse anti-BrdU (364104) (*BioLegend Inc.*). *Alexa fluor* secondary antibodies used in immunofluorescence microscopy experiments were from *Molecular Probes* (*Thermo Fisher Scientific*).

### Recombinant MBLAC1 production in *E. coli* and purification

cDNA (codon optimised for expression in *E. coli*) encoding for MBLAC1 (*GeneART, Thermo Fisher Scientific*) was inserted into the pCOLD I vector (*Addgene*) in order to produce MBLAC1 with an *N*-terminal hexa-histidine tag (6xHis) with an *N*-terminal C human rhinovirus (HRV3C) protease cleavage site. Recombinant MBLAC1 protein was produced in *E. coli* BL21 (DE3) cells grown in 2TY growth media with 50 µg/mL ampicillin at 37°C to mid-exponential phase (OD 600 = 0.6–0.8). MBLAC1 production was induced by addition of isopropyl β-D-1-thiogalactopyranoside (IPTG) (0.5 mM) supplemented with 50 µM zinc sulfate while incubating at 15°C. Cells were harvested by centrifugation (6500 x g, 8 min) after 18 hr and frozen in liquid nitrogen. The cell pellet (~20 g) was added to 100 mL of lysis buffer (20 mM 4-(2-hydroxyethyl)−1-piperazineethanesulfonic acid (HEPES), pH 7.5, 500 mM sodium chloride, 5 mM imidazole), lysed by sonication followed by centrifugation (20,000 x g, 20 min). MBLAC1 was purified by loading the supernatant onto a 5 mL Ni-affinity column (*GE Healthcare*). The equilibration buffer was the same as the lysis buffer. The elution buffer additionally contained 500 mM imidazole and was used to form an imidazole gradient (from 5 to 500 mM) to elute the His-tagged protein. MBLAC1 was then loaded onto a S200 (300 mL) gel filtration column using 20 mM HEPES, pH 7.5, 500 mM sodium chloride as the running buffer. Fractions containing protein were analysed by SDS-PAGE. The 6x-His tag was cleaved by addition of HRV3C protease (overnight incubation at 4°C), then purified by Ni-affinity chromatography to remove the cleaved tag. The resultant purified MBLAC1 (~27 kDa) was buffer exchanged into 25 mM HEPES, pH 7.5, 30 mM sodium chloride, then concentrated to 23 mg/mL using a Centricon concentrator (10 k MW cutoff) (*Merck*) (at 3000 x g) until the desired volume was achieved.

### Crystallization and structure determination

Crystallization was performed using the sitting drop vapor diffusion method using *Art Robbins* 96 well - three subwell Intelliplates and a protein:well drop size of 200 nL:100 nL,100 nL:100 nL, 100

nL:200 nL for each subwell condition. MBLAC1 crystallised over ~24 hr using the following conditions: JCSG-*plus* condition 94, 0.1 M bis Tris pH 5.5, 0.2 M ammonium acetate, 25 % w/v PEG 3350 (protein to reservoir ratio 2:1, 1:1, and 1:2) (*Molecular Dimensions*). Crystallization conditions were optimised by varying the ammonium acetate concentration (0.14–0.25 M) and PEG 3350 percentage (23–30%). The resultant crystals (~150×300 µm) obtained in 0.1 M Bis Tris pH 5.5, 0.18 M ammonium acetate, and 25 % w/v PEG 3350, were cryo-protected in well-solution diluted to 12.5% sucrose for 30 s, then harvested using nylon loops followed by cryo-cooling and storage under liquid nitrogen. Data were collected on a single crystal at 100 K at the Diamond Light Source Synchrotron (beamline I04, 1.28268 Å wavelength) to 1.8 Å resolution. Data were autoprocessed at the beamline using XDS (*Kabsch, 2010*) and CCP4-SCALA (*Evans, 2006*) in XIA2 (*Winter, 2010*). The MBLAC1 structure was solved by molecular replacement (MR) using the PHASER subroutine within PHENIX (*Read, 2001*; *Adams et al., 2004*; *Adams et al., 2010*) with an ensemble structure as a search model based on 11 crystal structures identified by the Phyre two modelling server (*Kelley et al., 2015*) using the MBLAC1 protein sequence as search input. Refinement was carried out by iterative rounds of model building using Coot (*Emsley et al., 2010*) and maximum likelihood restrained refinement using PHENIX (*Afonine et al., 2005*). Ramachandran statistics calculated 98.27% most favored geometry, 1.61% additionally allowed and 0.12% outliers. Data collection, processing, and structure refinement statistics are given in *Table 1* (related to *Figure 1*).

## Non-denaturing electrospray ionization mass spectrometry

MBLAC1 purified from *E. coli* was diluted to 15 µM in 15 mM ammonium acetate buffer (pH 7.5). The cone voltage for the acquisition of the spectra was 80 V. Electrospray ionization mass spectra of the *apo*-enzyme were acquired in the positive ionization mode after overnight metal removal treatment by addition of 20 mM ethylenediaminetetraacetic acid (EDTA) to the protein solution.

## Inductively coupled plasma mass spectrometry (ICP-MS)

Wild-type MBLAC1 was produced in *E. co*li or HEK293 cells; 30 µg of the resultant purified protein was then buffer exchanged into 20 mM HEPES pH 7.5, 50 mM NaCl buffer prepared using 'ultra-pure' water. Concentrations of Zn, Fe, Mn, Ni, and Co divalent ions were measured in triplicate. ICP-MS experiments and data analysis were carried out by the ICP-MS Trace Element Small Research Facility of the Earth Sciences Department, Oxford University.

## Cell culture and transfection

HeLa (*ATCC*) and HEK293 (kindly provided by Prof. Peter McHugh) cells were cultured in Dulbecco's modified Eagle's medium supplemented with 10% fetal bovine serum, and 2 mM L-glutamine, at 37°C in a humidified incubator (5% $CO_2$). To generate the construct MBLAC1 6xHis-pCDNA 3.1, an appropriate sequence coding for MBLAC1 (*GeneART*, *Thermo Fisher Scientific*) was amplified by PCR to introduce a C-terminal 6xHis tag, then cloned into the pCDNA 3.1 vector (*Invitrogen*). Both HeLa and HEK293 cells were transiently transfected with the MBLAC1 6xHis-pCDNA3.1 construct using the Fugene HD transfection reagent (*Promega*), following the manufacturer's instructions. MBLAC1 production was monitored by western blotting, 24 and 48 hr after transfection. To generate a stable cell line overexpressing MBLAC1, HEK293 cells were transfected with the MBLAC1 6xHis-pCDNA 3.1 construct using the Fugene HD transfection reagent (*Promega*), following the manufacturer's instructions. Stably transfected cells were selected by addition of geneticin (G418) (1 mg/ml) to the cell culture media; cell clones were generated by limiting dilution plating. Clones were analysed by western blotting for their capacity to produce MBLAC1.

## HEK293 expressed MBLAC1 production and purification

The best producing MBLAC1 HEK clone (c27) was adapted to serum-free growing conditions (*Free style* medium, *Invitrogen*) and cultivated in spinning culture at 37°C in a humidified incubator (8% $CO_2$) in the presence of 20 µM heparin sodium salt. Cell medium containing secreted 6xHis-tagged MBLAC1 was harvested every 48 hr and stored at −20°C until used. For purification, 10 mM imidazole was added to 1 L of clarified cell medium before loading onto a 1 mL Ni-affinity column (GE Healthcare) equilibrated with 25 mM HEPES, pH 7.5, 500 mM sodium chloride, 10 mM imidazole. The elution buffer additionally contained 500 mM imidazole and was used to form an imidazole

gradient (from 10 to 500 mM) to elute the His-tagged protein. Fractions containing protein were analysed by SDS-PAGE. The solution was concentrated to about 100 µL using a Centricon concentrator (10 k MW cutoff) (*Merck*) (at 3000 x g) and exchanged into 25 mM HEPES pH 7.5, 50 mM NaCl using 0.5 mL *Micro Biospin*<sup></sup>*six* desalting columns (*Biorad*). The obtained protein solution was concentrated to about 5 mg/mL and its purity determined by SDS PAGE. Alternatively, wild-type HEK293 cells adapted to serum-free growing conditions (*Free style* medium, *Invitrogen*) were transiently transfected with the MBLAC1 6xHis-pCDNA 3.1 encoding constructs (for the expression of both wild-type and active site mutant MBLAC1) using the Expi293 expression system (*Thermo Fisher Scientific*), following the manufacturer's instructions. Interestingly, mutMBLAC1, unlike wtMBLAC1, was not secreted into the cell medium. Therefore both wild-type and active site mutant MBLAC1 were purified as previously described following cell lysis by sonication.

## Single point mutagenesis

Site-directed residue substitutions of MBLAC1 were performed using *Pfu* Turbo DNA polymerase (*Agilent Technologies*) according to the manufacturer's protocol. Primers used to generate the mutants were designed following the manufacturer instructions and are listed in *Table 2*. The pCOLD I vector (used to overexpress MBLAC1 in *E. coli*) was employed as template DNA. Plasmid DNA with the correct mutation were used to transform *E. coli* BL21 (DE3) cells as described above. MBLAC1 mutants expressed in *E. coli* were produced and purified similarly to wild-type enzyme, as previously described.

## Subcellular fractionation

Cytoplasmic and nuclear subcellular fractions were prepared as follows: HeLa cells were washed twice with ice-cold PBS, then suspended in harvest buffer (10 mM HEPES KOH pH 7.9, 50 mM NaCl, 0.5 M sucrose, 0.5% Triton X-100, 1 mM DTT and protease inhibitor cocktail). After incubation for 5 min on ice, cells were centrifuged (100 x g, 10 min, 4°C) to pellet nuclei. The supernatant (containing cytosolic and membrane proteins) was centrifuged (14,000 x g, 10 min, 4°C) and stored. The nuclear pellet was resuspended and washed by centrifugation in buffer A (10 mM HEPES KOH pH 7.9, 10 mM KCl, 1 mM DTT and protease inhibitor cocktail) (100 x g10 min, 4°C). The obtained nuclear pellet was resuspended in 4 volumes of buffer C (10 mM HEPES KOH pH 7.9, 200 mM NaCl, 0.1 % NP-40, 1 mM DTT and protease inhibitor cocktail) and vortexed at 4°C for 30 min. The nuclear lysate was then centrifuged (15 min at 14,000 x g at 4°C). Proteins concentrations of both cytoplasmic and nuclear fractions were measured using the Bradford reagent. The subcellular fractions were then evaluated by western blot analysis.

## In vitro RNA degradation assays

DNA sequences coding for a portion of the human histone 2H3C (HIST2H3C) (350 nt), and the WT and MUT shorter forms of the human histone 2H3C (HIST2H3C) (168 nt) genes with a 5' T7 RNA polymerase promoter sequence were used as templates to generate internally [$^{32}$P] labeled RNA substrates by in vitro transcription (MEGAscript T7 Kit, *Thermo Fisher Scientific*). The reaction was carried out following the manufacturer's instructions by adding an excess of [α-$^{32}$P] UTP to the reaction mixture. In vitro transcription products were fractionated on a 6% denaturing polyacrylamide/urea gel. The generation of the radiolabeled RNA substrates was monitored by autoradiography. The radiolabeled RNA was then gel purified and resuspended in 80 µL ddH$_2$O. For the 5' [γ-$^{32}$P] ATP labeled substrates listed in *Table 1*, RNA fragments were purchased from *Integrated DNA Technologies* and 1 µg of each RNA was treated with T4 polynucleotide kinase (*New England Biolabs*) in the presence of an excess of [γ-$^{32}$P] ATP. The obtained 5' [γ-$^{32}$P] ATP labeled substrates were then gel purified using a 10% denaturing polyacrylamide/urea gel and resuspended in 60 µL ddH$_2$O. Cleavage assays were carried out in a mixture (10 µl) containing: 1 µL of labeled RNA substrates (~600 ng of [α-$^{32}$P] UTP substrate (~5 µM),~16 ng of [γ-$^{32}$P] ATP fragments (~1 µM)), 500 ng (~15 µM) of recombinant wild-type or the active site variant MBLAC1 produced in both *E. coli* and HEK293 cells, or 50 ng (~1.5 µM) of recombinant wild-type or the active site variant produced in HEK293 cells and 1x reaction buffer (20 mM HEPES KOH pH 7.5, 50 mM KCl, 10 mM MgCl$_2$, 0.5 mM DTT, 0.05% Triton X-100, 5% glycerol) at 37°C. The reaction was stopped at the appropriate time point by phenol-chloroform extraction and ethanol precipitation. RNA degradation products were then

**Table 2.** Oligonucleotide primers and single guide RNA targeting sequences used in site directed mutagenesis, RT-qPCR experiments and CRISPR/CAS9 mediated KD studies.

Abbreviations: GB, gene body; RT, read through. Note that the standard BBL numbering system has been used (*Galleni et al., 2001*).

| Oligonucleotide primers used for site directed mutagenesis | |
| --- | --- |
| *E. coli* produced HSE | |
| H196A_F | GCAACACCGGGTGCTGGTGGTCAGCG |
| H196A_R | CGCTGACCACCAGCACCCGGTGTTGC |
| D221A_F | GTTGTTGCCGGTGCTGTTTTTGAACGTG |
| D221A_R | CACGTTCAAAAACAGCACCGGCAACAAC |
| H263A_F | GGTTGTTCCTGGTGCTGGTCCGCCTTTTCG |
| H263A_R | CGAAAAGGCGGACCAGCACCAGGAACAACC |
| Oligonucleotide primers used in RT-qPCR experiments | |
| GAPDH_GB_F (intron) | ACCCAGAAGACTGTGGATGG |
| GAPDH_GB _R (intron) | TTCAGCTCAGGGATGACCTT |
| GAPDH _RT_F | TCCAGCCTAGGCAACAGAGT |
| GAPDH _RT_R | TGTGCACTTTGGTGTCACTG |
| HIST1H2BC_GB_F | ACCTCCAGGGAGATCCAGAC |
| HIST1H2BC_GB_R | AGCTGGTGTACTTGGTGACG |
| HIST1H2BC_RT_F | CTCCAGGGAGATCCAGACGG |
| HIST1H2BC_RT_R | GCTCTTTTAGTGGGTATCTGGG |
| HIST4H4_GB_F | GAAGGTGCTGCGGGACAATA |
| HIST4H4_GB_R | AAGACTTTGAGGACTCCCCG |
| HIST4H4_RT_F | CGCGCAACGCAGTAGTGACC |
| HIST4H4_RT_R | CATTCAGCTTTCGGGCTTGCAGGTA |
| HIST2H2AC_GB_F | GGCTCGGGACAACAAGAAGA |
| HIST2H2AC_GB_R | AGAACGGCCTGGATGTTAGG |
| HIST2H2AC_RT_F | GAAAGCCACAAAGCCAAAAGC |
| HIST2H2AC_RT_R | GGCTTGACACCATACTCATTCACC |
| HIST1H3G_GB_F | CTGAGCTGCTGATCCGCAAG |
| HIST1H3G_GB_R | GGCGAGCGAGCTGAATGTCC |
| GAPDH_GB_F (exon-exon) | AAGGTGAAGGTCGGAGTCAA |
| GAPDH_GB _R (exon-exon) | AATGAAGGGGTCATTGATGG |
| r18S_F | GGCCCTGTAATTGGAATGAGTC |
| r18S_R | CCAAGATCCAACTACGAGCTT |
| CRISPR/Cas9 single guide RNA (sgRNA) | |
| Location | sgRNA targeting sequences |
| 5' targeting | ACAGCGGCTCGGTCCGCATGAGG |
| 3' targeting | TGCACTAATCAGCCTCGAGAGGG |
| CRISPR/Cas9 sequencing primers | |
| | Primers sequence |
| CRISPR/Cas9_F | CAGAGAACCGAGGCTTAGGG |
| CRISPR/Cas9_R | GAAGCTCCCACCCTTGACTG |
| CRISPR/Cas9 RT-qPCR primers | |
| | Primers sequence |
| HSE_set1_F | ATCACATCGGGAACTTGGGG |
| HSE_set1_R | CACCACGGTGCCCAGAG |

*Table 2 continued on next page*

*Table 2 continued*

**Oligonucleotide primers used for site directed mutagenesis**

| *E. coli* produced HSE | |
| --- | --- |
| HSE_set2_F | GCTCTGGGCACCGTGG |
| HSE_set2_R | GCGAGGCTTCCCTTAACACT |

DOI: https://doi.org/10.7554/eLife.39865.016

fractionated on 6, 8 or 10% denaturing polyacrylamide/urea gel and the data were analysed by Phospho-Imager (Fujifilm FLA 5000 imager).

## siRNA transfection and cell synchronization

About 50% confluent HeLa cells were transfected with a SMARTpool of siRNA for MBLAC1 or CPSF73 (*GE Dharmacon*), or with a negative control siRNA (Luciferase), using the Lipofectamine RNAiMAX transfection reagent (*Invitrogen*) following the manufacturer's instructions. Cells were transfected with a final concentration of 10 nM siRNA and knockdown efficiency was assessed 48 hr post transfection by western blot analysis. Transfected HeLa cells were synchronised using the double thymidine block method (*Harper, 2005*). In brief, 6 hr post-siRNA transfection, cells were treated with 2 mM thymidine (final concentration) for 18 hr; the thymidine was then removed for 9 hr, and then was added again (2 mM) for 15 hr. Cells were then washed with PBS twice, and harvested at different time points after release from the block; alternatively, cells were washed with PBS twice and treated with 10 µM of 5-bromo-2'-deoxyuridine (BrdU), for 30 min or 1 hr. BrdU treated cells were washed with PBS to remove the unincorporated BrdU and then harvested at different time points for cell cycle analysis. Similarly, wt and CRISPR/cas9 mediated MBLAC1 kd HeLa cells were synchronised and BrdU treated as described above.

## Cell cycle analysis

Synchronised HeLa cells were washed with PBS buffer and fixed in ice-cold ethanol overnight. DNA was stained with a propidium iodide (PI) solution containing 0.1% Triton X-100 in PBS, 0.2 mg/ml RNase A, 0.02 mg/ml PI for 30 min at room temperature. Alternatively, BrdU-treated cells were fixed by adding ice-cold 70% ethanol overnight. Cells were then incubated with 2 N HCl, 0.5% Triton X-100 for 30 min at room temperature and washed with 0.1 M sodium tetraborate buffer, pH 8 for 2 min. Cells were washed and incubated with a FITC conjugated anti-BrdU antibody (*BioLegend*) for 1 hr at room temperature. PI staining followed as described above. Cell cycle profiles were analysed by flow cytometry with FACSCalibur (*BD Biosciences*) and processed using FlowJo software (*Tree Star*).

## Chromatin RNA extraction and libraries preparation

Chromatin RNA (ChrRNA) was extracted from HeLa cells 48 hr after siRNA transfection to coincide with the end of the second thymidine block time-point. Approximately $3 \times 10^6$ cells were resuspended in 500 µL of ice-cold RLB buffer (10 mM Tris-HCl, pH 7.5, 140 mM NaCl, 0.5% nonidet-P40 (NP-40), 1.5 mM MgCl$_2$) and lysed by adding an equal volume of RLB buffer with 24% (m/v) sucrose. Nuclei were collected by centrifugation (14,000 x g) at 4°C for 10 min. Isolated nuclei were resuspended in 120 µl of NUN1 buffer (20 mM Tris-HCl, pH 7.9, 75 mM NaCl, 0.5 mM EDTA, 50% Glycerol, 0.125 mM PMSF, 1 mM DTT), followed by addition of 1.2 ml NUN2 buffer (20 mM HEPES KOH pH 7.6, 7.5 mM MgCl$_2$, 0.2 mM EDTA, 300 mM NaCl, 1 M urea, 1 % NP-40, 1 mM DTT). Nuclei were incubated for 15 min on ice with mixing by vortexing for 5 s every 5 min. Chromatin pellets were precipitated by centrifugation (14,000 x g) at 4°C for 10 min and then resuspended in 200 ml HSB buffer (10 mM Tris-HCl, pH 7.5, 500 mM NaCl, 10 mM MgCl$_2$) in presence of 4 U TURBO DNase (*Ambion*) and incubated at 37°C for 20 min. Following a 20-min proteinase K treatment (*Roche*) at 37°C, chromatin RNA (chrRNA) was phenol-chloroform extracted and ethanol precipitated. RNA was resuspended in 200 µL of 1x TURBO DNAse buffer in presence of 4 U TURBO DNase and incubated at 37°C for 30 min. The entire procedure of extraction, precipitation and DNase treatment was

repeated twice. RNA was then extracted and precipitated again, prior to solubilisation in 75 µL of RNase free water.

Prior to RNA libraries preparation, rRNA was depleted using the Ribo-Zero Magnetic kit (*Illumina*) from 2.5 µg of ChrRNA. Libraries were prepared starting from 100 ng chrRNA using the NEBNext Ultra Directional RNA Library Prep kit for Illumina (*NEB*) following the manufacturer's instructions. Deep sequencing using Hiseq4000 with paired-end (75 bp) runs was performed by the Wellcome Trust Centre for Human Genetics (WTCHG), Oxford, UK.

## Real time PCR analyses

ChrRNA was extracted from synchronised HeLa cells 48 hr after siRNA transfection as described above. For reverse transcription, 500 to 900 ng of ChrRNA from each sample was used to generate single-strand cDNA by incubation with random hexamers (*Qiagen*) and Superscript III reverse transcriptase (*Life Technologies*). Quantitative RT-PCR was performed using the SYBR Green Master Mix (*Qiagen*). Relative RNA levels were calculated using the ΔCt method. Data were acquired and analysed using Rotor-GeneQ (*Qiagen*). Defects in 3' end transcription termination were evaluated using amplicons located downstream of the 3' untranslated region (3' UTR) of RD histone genes. The relative abundance of unprocessed sequences was normalised to the relative abundance of an amplicon covering a portion of the GAPDH gene body (GB) or to the corresponding histone GB abundance. Primers used are listed in *Table 2*. Similarly, chrRNA was extracted from synchronised wild-type and CRISPR/Cas9-mediated MBLAC1-depleted HeLa cells; cDNA libraries and quantitative RT-PCR were performed as above.

## Chromatin RNA-Seq data processing

GRCh38 was used as a reference genome. RNA-sequencing reads were trimmed using Cutadapt 1.8.3 (*Martin, 2011*). An in-house Perl scripts Perl script was used to remove the reads left unpaired. The remaining reads were then aligned to the human reference genome with Tophat 2.0.13 (*Sullivan et al., 2009b*; *Sun et al., 2006*) using parameters: tophat -g 1 r 3000 –no-coverage-search. Aligned reads were processed to only include properly paired, properly mapped reads with no more than two mismatches using SAMtools 1.2 (*Li et al., 2009*). Data were scaled to library size (genome-CoverageBed) using Bedtools (*Quinlan and Hall, 2010*). For data vizualization, trackhubs for the UCSC browser were created by employing the UCSC bedGraphToBigWig tool (*Kent et al., 2002*). For the gene body normalised plots, expression levels (as reads per kilobase per million mapped reads, RPKMs) were calculated for a region of 'TES-100bp' for each gene in all the three samples, namely, LUC, MBLAC1 and CPSF73. These values were then used to normalise each base position followed by plotting them using Python script. Gene body normalised plots were then log two transformed.

## RNA-chromatin immunoprecipitation analysis

Wild-type and MBLAC1 C-terminal FLAG knock in HeLa cells were synchronised in early S-phase using a double thymidine block and used in RNA-chromatin immunoprecipitation (RNA-ChIP) experiments as described by B. K. Sun and J. T. Lee (*Sun et al., 2006*). An additional TURBO DNase (*Ambion*) treatment was repeated at the end of the procedure to ensure complete chromatin digestion. The obtained immunoprecipitated RNA was used to prepare cDNA libraries and quantitative RT-PCR was performed as above. RNA levels were calculated using the standard curve method. Data were acquired and analysed using Rotor-GeneQ (*Qiagen*). Primer sets used in the experiments are listed in *Table 2*.

## CRISPR/Cas9-mediated MBLAC1 deletion

Single guide RNA (sgRNA) constructs aiming to delete the MBLAC1 gene in HeLa cells were designed and produced by the Genome Engineering Oxford Facility, Oxford University, UK (*Table 2*). HeLa cells were transiently co-transfected with two sgRNA constructs (vector epX459(1.1)), both carrying the engineered hSpCas9(1.1) gene (*Slaymaker et al., 2016*) and a sgRNA cassette targeting the 5' or 3' region flanking the MBLAC1 gene using Fugene HD transfection reagent (*Promega*), following the manufacturer's instructions. 24 hr after transfection, cells were exposed to puromycin (2 µg/ml) for 24 hr to positively select transfected cells. After recovery, cells were harvested and used

to select MBLAC1-depleted cell clones by limiting dilution plating and to isolate genomic DNA to evaluate the gene deletion efficiency by PCR (*Table 2*). Genomic DNA was then extracted from cell clones and PCR reactions amplifying the DNA region flanking the sgRNA targeting sites were performed to confirm MBLAC1 gene deletion. PCR products corresponding to the wt gene or to the gene deletion were gel purified and their sequence confirmed by Sanger sequencing. Total RNA was then extracted from cell clones carrying the MBLAC1 deletion using TRI reagent (*Ambion*) following the manufacturer's instructions. RT-qPCR analyses were performed to evaluate MBLAC1 mRNA levels after CRISPR/Cas9-mediated depletion as above. RT-qPCR products were analysed by Sanger sequencing.

### CRISPR/Cas9-mediated MBLAC1 C-terminal FLAG knock in generation

HeLa cells were transiently co-transfected with a single guide RNA (sgRNA) construct targeting the 3' region flanking the MBLAC1 gene (previously used to delete the MBLAC1 gene in HeLa cells) and with a pUC19 vector (*NEB*) carrying a FLAG tag sequence flanked at both 5' and 3' by ~300 nucleotides sequence corresponding to the genomic sequence flanking the 3' region of the MBLAC1 gene using Fugene HD transfection reagent (*Promega*), following the manufacturer's instructions. Similarly to the CRISPR/Cas9-mediated MBLAC1 deletion, 24 hr after transfection, cells were exposed to puromycin (2 µg/ml) for 24 hr to positively select transfected cells. After recovery, cells were harvested and used to select MBLAC1 knock in (Kin) cell clones by limiting dilution plating and to isolate genomic DNA to evaluate the FLAG sequence insertion efficiency by PCR (*Table 2*). Positive cells clones were growth as usual and analysed by western blot to validate the FLAG tag insertion at protein level using an anti-FLAG antibody.

### Polyadenylated RNA selection

Total RNA was extracted from wild-type and CRISPR/Cas9-mediated MBLAC1-depleted HeLa cells after synchronization in early S-phase using the TRI reagent (*Ambion*). The Dynabeads mRNA purification kit (*Ambion*) was used to positively select polyadenylated RNA following the manufacturer's instructions; unbound RNA was considered as the unpolyadenylated RNA fraction. Both RNA fractions were phenol-chloroform extracted and ethanol precipitated, then solubilized in 20 µL ddH$_2$O. For reverse transcription, 1 µg of total, polyadenylated and unpolyadenylated RNA from each sample was used to generate single-strand cDNA by incubation with random hexamers (*Qiagen*) and Superscript III reverse transcriptase (*Life Technologies*). Quantitative RT-PCR was performed using the SYBR Green Master Mix (*Qiagen*). RNA levels were calculated using the standard curve method. Data were acquired and analysed using Rotor-GeneQ (*Qiagen*). Primer sets used in the experiments are listed in *Table 2*).

### Data analysis and statistics

The number of biological repeats and statistical tests (conducted in Microsoft Excel or GraphPad) are indicated in the corresponding figure legends. Error bars representing s.e.m. are shown where appropriate.

## Acknowledgements

We thank the Medical Research Council, Cancer Research U.K., the Biotechnological and Biological Research Council, and the Wellcome Trust for funding. PG is supported by a Sir Henry Dale Fellowship jointly funded by the Wellcome Trust and the Royal Society [200473/Z/16/Z]. CRdeA was supported by a EMBO Long-Term Fellowship [ALTF 1351–2011]. NJP lab is funded by a European Research Council advanced grant [339270] and Wellcome Trust Investigator Award [107928/Z/15/Z]. We thank the Diamond Light Source Synchrotron, Didcot, UK; the Biophysical Services of the Biochemistry Dept., Oxford University, UK; the ICP-MS Trace Element Small Research Facility of the Earth Sciences Department, Oxford University, UK; the Wellcome Trust Centre for Human Genetics (WTCHG), Oxford, UK.

## Additional information

### Competing interests

Nick J Proudfoot: Reviewing editor, *eLife*. The other authors declare that no competing interests exist.

### Funding

| Funder | Grant reference number | Author |
|---|---|---|
| Wellcome Trust | 105045/Z/14/Z | Ilaria Pettinati |
| Wellcome Trust | Sir Henry Dale Fellowship 200473/Z/16/Z | Pawel Grzechnik |
| Royal Society | Sir Henry Dale Fellowship 200473/Z/16/Z | Pawel Grzechnik |
| European Molecular Biology Organization | Long-Term Fellowship ALTF 1351–2011 | Claudia Ribeiro de Almeida |
| Wellcome Trust | 107928/Z/15/Z | Nick J Proudfoot |
| European Research Council | 339270 | Nick J Proudfoot |
| Medical Research Council | | Christopher J Schofield |
| Wellcome Trust | 105045/Z/14/Z | Christopher J Schofield |
| Cancer Research UK | | Christopher J Schofield |
| Biotechnology and Biological Sciences Research Council | | Christopher J Schofield |

The funders had no role in study design, data collection and interpretation, or the decision to submit the work for publication.

### Author contributions

Ilaria Pettinati, Conceptualization, Data curation, Formal analysis, Validation, Investigation, Visualization, Methodology, Writing—original draft, Writing—review and editing, Designed, performed, and analysed the experiments, Wrote the manuscript; Pawel Grzechnik, Methodology, Advised on in vitro assays and RNA-seq data analysis; Claudia Ribeiro de Almeida, Methodology, Advised on cell-based experiments, Analysed flow cytometry samples; Jurgen Brem, Methodology, Advised on biochemical analyses; Michael A McDonough, Methodology, Advised on crystallography; Somdutta Dhir, Formal analysis, Validation, Performed the bioinformatics analyses; Nick J Proudfoot, Supervision, Writing—review and editing, Supported and advised on the molecular biology and genetic parts of the project, Wrote the manuscript; Christopher J Schofield, Conceptualization, Supervision, Writing—review and editing, Conceived the project, Analysed data, Wrote the manuscript

### Author ORCIDs

Ilaria Pettinati (iD) https://orcid.org/0000-0002-5904-7984
Pawel Grzechnik (iD) https://orcid.org/0000-0003-3355-1741
Michael A McDonough (iD) https://orcid.org/0000-0003-4664-6942
Somdutta Dhir (iD) https://orcid.org/0000-0003-1987-3971
Nick J Proudfoot (iD) https://orcid.org/0000-0001-8646-3222
Christopher J Schofield (iD) https://orcid.org/0000-0002-0290-6565

### Decision letter and Author response

Decision letter https://doi.org/10.7554/eLife.39865.024
Author response https://doi.org/10.7554/eLife.39865.025

## Additional files

### Supplementary files
• Transparent reporting form
DOI: https://doi.org/10.7554/eLife.39865.017

### Data availability
Diffraction data have been deposited in PDB under the accession code 4V0H. ChrRNA-seq data generated during this study have been deposited in the Gene Expression Omnibus under accession code GSE94686.

The following datasets were generated:

| Author(s) | Year | Dataset title | Dataset URL | Database and Identifier |
|---|---|---|---|---|
| McDonough MA, Brem J, Schofield CJ, Pettinati I | 2018 | Human metallo beta lactamase domain containing protein 1 (hMBLAC1) | https://www.rcsb.org/structure/4V0H | Protein Data Bank, 4V0H |
| Pettinati I, Ribeiro de Almeida C, Dhir S, Proudfoot NJ, Schofield CJ | 2018 | Biosynthesis of histone messenger RNA employs a specific 3′ end endonuclease | https://www.ncbi.nlm.nih.gov/geo/query/acc.cgi?acc=GSE94686 | Gene Expression Omnibus, GSE94686 |

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
