## [Decision Letter]

Thank you for submitting your article "Biosynthesis of histone messenger RNA employs a specific 3' end endonuclease" for consideration by *eLife*. Your article has been reviewed by three peer reviewers, and the evaluation has been overseen by a Reviewing Editor and James Manley as the Senior Editor. The reviewers have opted to remain anonymous.

The reviewers have discussed the reviews with one another and the Reviewing Editor has drafted this decision to help you prepare a revised submission. Given the significance of the points raised, the paper will need to be seen again by the reviewers.

The paper by Schofield, Proudfoot, and colleagues characterizes the MBLAC1 protein. The authors obtain a crystal structure of the protein, which suggests that MBLAC1 could function as an RNase. Biochemical assays provide largely convincing evidence that MBLAC1 indeed possesses endonuclease activity and has a preference for CA sequence. To investigate its in vivo function, the authors knock down MBLAC1 and observe cell cycle and transcription termination defects, which are suggested to be specific for replication-dependent (RD) histone genes.

Overall the structural and biochemical data convincingly show that MBLAC1 can function as an RNA endonuclease. However, the reviewers are not fully convinced that MBLAC1 is specifically involved in RD histone RNA 3' end processing and suggest additional analyses to examine this key question.

Essential revisions:

1) The meta-gene analyses show overall higher expression levels for histone genes both upstream and downstream of the transcript end site (TES) (Figure 4D). Hence, the read-through signal appears much less significant than what is stated. This is unusual for a termination factor. Could the higher signals downstream of the TES simply be an indirect effect of higher histone mRNA expression or is this a normalization issue? If the latter, this data should be presented as gene-body normalized and/or as heat maps illustrating all individual histone mRNAs. Also, the genome browser images (e.g. Figure 4B-C) do not fully convey how the gene-body vs read-through signal relate to one another.

2) RD histone RNA 3'end processing has been extensively studied and the machinery is well characterized. The authors fail to provide strong evidence linking MBLAC1 to known RD histone RNA 3'end processing factors. They cite that MBLAC1 binds to Clp1, but it is not clear that Clp1 functions in RD histone RNA processing. Is it possible that MBLAC1 might participate independent of other known factors? To demonstrate this, the authors need to provide evidence that MBLAC1 at least associates with histone gene TESs, for example by ChIP assays.

3) Figure 1, Figure 5 and Figure 6: The authors should show the purity of the purified proteins on silver-stained SDS gels. In order to completely exclude contaminating nuclease activity, results in Figure 5 and Figure 6 could be complemented with catalytically dead protein purified from HEK293 cells.

4) Figure 3, Figure 4, Figure 4—figure supplement 1, Figure 4—figure supplement 2: How is it possible that histone H3 protein levels are down upon MBLAC1 depletion (either by siRNA or CRISPR), when the corresponding mature mRNA levels appear either unaffected or increased?

5) The data presented in the paper doesn't convincingly show that MBLAC1 is selective for RD histone transcripts: In Figure 4B/ Figure 4—figure supplement 2A, there are more reads immediately downstream of the TES in siMBLAC1 than in the siLuc control. Could there also be a defect in processing of polyadenylated transcripts? It would be helpful to show a metagene analysis for canonical mRNAs (as in Figure 4D).

[Editors' note: further revisions were requested prior to acceptance, as described below.]

Thank you for resubmitting your work entitled "Biosynthesis of histone messenger RNA employs a specific 3' end endonuclease" for further consideration at *eLife*. Your revised article has been favorably evaluated by James Manley (Senior Editor) and a Reviewing Editor.

The manuscript has been improved but there are some remaining issues that need to be addressed before acceptance, as outlined below:

1) Insufficient labeling of figures is a source of confusion and should be improved.

2) The CLP1 co-IP still appears misplaced. Consider removing it entirely.

3) Figure 4—figure supplement 3: In the interest of clarity, how about plotting the three tracks (siLuc, siCPSF73, siMBLAC1) in the same graph? Try a log2-transformation of the data to make the read-through region stand out more clearly?

4) Discussion section: “[…] recognition of the secondary structure of the stem-loop by MBLAC1 is important in maintaining the specificity of the cleavage” Why recognition? Is it not more likely that the structure prevents MBLAC1 to access the putative single-stranded cleavage sites?

5) The MBLAC1 RNA-ChIP experiment is not convincing. At a very minimum the authors should include an input sample to assay RDH expression relative to GAPDH and to confirm that RDH expression is not affected by the tagging of MBLAC1.

[Editors' note: further revisions were requested prior to acceptance, as described below.]

Thank you for resubmitting your work entitled "Biosynthesis of histone messenger RNA employs a specific 3' end endonuclease" for further consideration at eLife. Your revised article has been favorably evaluated by a Reviewing Editor and a Senior Editor.

The manuscript has been improved but there are some remaining issues that need to be addressed before acceptance, as outlined below:

The authors have addressed most of the concerns appropriately. However, one concern remains regarding the generality of MBLAC1 in RD histone termination. This should be clarified both in the figures and in the text. Especially the RD histone tracks in Figure 4—figure supplement 3 are not convincing to state that MBLAC1 is generally involved in this process.

Specific comments concerning Figure 4—figure supplement 3:

1) The interpretation of the shown plots is left as an exercise to the reader. To resolve this, 2 modifications can be suggested: 1) Indicate clearly in each plot the genomic region where an increased read-through upon knock-down of MBLAC1 and/or CPSF73 is observed, and 2) Sort or group plots into classes where read-through is observed, not observed or ambiguous. Once, this is done, please modify the text accordingly (it looks like only a subset of RD histone genes display read-through upon MBLAC1 knock-down).

2) The claimed gene-body normalization in panel A is not apparent from the shown data. Why?

3) The figure legend claims that regions (TSS-100 bp, TES+200 bp in panel A) and (TSS-200 bp, TES +200 bp in panel B) are depicted. Neither appears to reflect what is shown. Is it regions TES-100 to TES+200?

4) Are the data showing the average of the 2 replicates or something else? This is not clarified.

5) Why should log2 transformation not be compatible with gene-body normalization of the data?

---

## [Author Response]

Essential revisions:1) The meta-gene analyses show overall higher expression levels for histone genes both upstream and downstream of the transcript end site (TES) (Figure 4D). Hence, the read-through signal appears much less significant than what is stated. This is unusual for a termination factor. Could the higher signals downstream of the TES simply be an indirect effect of higher histone mRNA expression or is this a normalization issue? If the latter, this data should be presented as gene-body normalized and/or as heat maps illustrating all individual histone mRNAs. Also, the genome browser images (e.g. Figure 4B-C) do not fully convey how the gene-body vs read-through signal relate to one another.

We thank the reviewer for these very helpful comments and suggestions. We understand the concern about the meta-gene analyses. Unfortunately, the higher histone gene expression level observed in the control sample (siLUC) leads to a normalization issue in comparing the extent of read through with the knock downs when the data were scaled to library size (which is a standard method for meta-gene analyses). To avoid the normalization issue the raw data were plotted for individual histone genes (Figure 4—figure supplement 3) showing in more detail the different histone genes that are affected by MBLAC1 or CPSF73 knock down. This approach enables better comparison with the results of other experiments where the relative abundance of unprocessed RNA was normalized not only to the abundance of a house-keeping gene (GAPDH), but also to that of the corresponding histone gene-body (as suggested by the reviewer). The results from the two methods are consistent (Figure 4D,E). We believe that the revaluation of these data will reassure the reviewer regarding the authentic effect of MBLAC1 on histone pre-mRNA.

2) RD histone RNA 3'end processing has been extensively studied and the machinery is well characterized. The authors fail to provide strong evidence linking MBLAC1 to known RD histone RNA 3'end processing factors. They cite that MBLAC1 binds to Clp1, but it is not clear that Clp1 functions in RD histone RNA processing. Is it possible that MBLAC1 might participate independent of other known factors? To demonstrate this, the authors need to provide evidence that MBLAC1 at least associates with histone gene TESs, for example by ChIP assays.

We thank the reviewer for these comments and suggestions for helpful new experiments, which we have carried out. We agree with the reviewer’s suggestion that MBLAC1 may well act independently from other known histone pre-mRNA processing factors, as we suggest in the manuscript and as supported by the observation MBLAC1 manifests substrate specificity in its purified form (see Figure 5 and Figure 6). We have carried out ChIP analyses to investigate the potential association of MBLAC1 with histone genes. Unfortunately, although our results pointed in this direction, the background signal was too high to allow definitive analyses. We therefore focused our efforts on RNA-Chromatin immunoprecipitation analyses. In both cases (ChIP and RNA-ChIP experiments) we used a C-terminal FLAG tagged MBLAC1 cell line generated from HeLa cells using CRISPR/Cas9 technology. This enabled us to use an anti-FLAG antibody to immunoprecipitate MBLAC1 at endogenous levels (anti-MBLAC1 antibodies suitable for IP are not currently commercially available). As shown in Figure 4G the comparison of controls (from both WT HeLa cells and FLAG cell line) and anti-FLAG samples (from FLAG tagged MBLAC1 knock in cells) clearly shows that the observed RIP signal for histone mRNA is not due to background but is MBLAC1 specific. This proposal is also supported by comparison of the signal detected for RD histones and normal protein coding mRNA (GAPDH) obtained after immunoprecipitation of MBLAC1 or FLASH. As shown, both anti-FLAG and anti-FLASH samples result in very similar RNA-ChIP profiles. We used FLASH as internal control because it is a well-known member of the 3′ histone pre-mRNA machinery (Marzluff, 2017). Even though the resolution limit of the technique cannot clearly distinguish between the TSS and TES of the histone mRNA due to the small size of these transcripts, the similar enrichments shown by both anti-FLAG and anti-FLASH are reassuring.

3) Figure 1, Figure 5 and Figure 6: The authors should show the purity of the purified proteins on silver-stained SDS gels. In order to completely exclude contaminating nuclease activity, results in Figure 5 and Figure 6 could be complemented with catalytically dead protein purified from HEK293 cells.

Following the reviewer’s suggestion, we now show protein purity by silver-stained SDS gel using wild-type and active site variant (mutMBLAC1) produced in both *E. coli* and HEK293 cells (Figure 5A). As shown, all the preparations are >95% pure (by gel analysis) except for the mutant enzyme produced in HEK293 cells. The somewhat lower HEK293 mutant (mutMBLAC1) enzyme purity implies a lack of contaminating nucleases: the observation mutMBLAC1 is less pure and less active is supportive of a lack of contaminating nucleases in our preparations (Figure 5A). The results shown in the previous versions of Figure 5 and Figure 6 have been complemented with new results (Figure 5D and Figure 6B); notably, analogous wild-type and active site variant (mutMBLAC1) proteins produced in HEK293 cells show similar catalytic properties to those produced in bacteria.

4) Figure 3, Figure 4, Figure 4—figure supplement 1, Figure 4—figure supplement 2 : How is it possible that histone H3 protein levels are down upon MBLAC1 depletion (either by siRNA or CRISPR), when the corresponding mature mRNA levels appear either unaffected or increased?

We thank the reviewer for the comment. These observations can be explained either by a compensating effect (low protein levels stimulate higher transcription or increased RNA stability) (Radhakrishnan, 2016), or by the increased stability and nuclear retention of unproperly terminated (possibly polyadenylated) RD histone pre-mRNA following incorrect 3′ end cleavage (Romeo, 2014). Either way, when the reviewer refers to “mature mRNA levels”, we believe that these transcripts should be considered as the totality of both properly and improperly processed histone pre-mRNA (this is because not all the improperly processed pre-mRNA appear to undergo polyadenylation).

5) The data presented in the paper doesn't convincingly show that MBLAC1 is selective for RD histone transcripts: In Figure 4B/ Figure 4—figure supplement 2A, there are more reads immediately downstream of the TES in siMBLAC1 than in the siLuc control. Could there also be a defect in processing of polyadenylated transcripts? It would be helpful to show a metagene analysis for canonical mRNAs (as in Figure 4D).

We thank the reviewer for this comment. In Author response image 1 we show the results of a meta-gene analysis, indicating that MBLAC1 depletion does not affect (within detection limits) canonical mRNAs.

[Editors' note: further revisions were requested prior to acceptance, as described below.]

The manuscript has been improved but there are some remaining issues that need to be addressed before acceptance, as outlined below:1) Insufficient labeling of figures is a source of confusion and should be improved.

We have worked to improve the labelling quality and the overall symmetry of each figure. We hope that now the figures appear also more consistent – please let us know if anything else needs attention.

2) The CLP1 co-IP still appears misplaced. Consider removing it entirely.

Following your suggestion, the CLP1 co-IP experiment has been removed. We understand the motivation for removing it and aim to carry out further investigations on the CLP1 interaction in the future.

3) Figure 4—figure supplement 3: In the interest of clarity, how about plotting the three tracks (siLuc, siCPSF73, siMBLAC1) in the same graph? Try a log2-transformation of the data to make the read-through region stand out more clearly?

Thank you for this suggestion. The expression levels of RD histone genes in the control sample (siLuc) are higher compared to those in both the MBLAC1 and CPSF73 depletions rendering the log2-transformation inappropriate for clear presentation. Thus, to address the issue, we have used gene-body normalization as follows: expression levels (as reads per kilobase per million mapped reads, RPKMs) were calculated for a region of 'TES-100bp' for each of the analysed histone genes in all the samples (siLUC, siMBLAC1, and siCPSF73). These values were then used to normalize each base position. For clarity, we have presented the data in both ways, i.e. (A) Gene-body normalization for each analysed histone gene and (B) Raw data plotted for individual histone genes (previous Figure 4—figure supplement 3).

4) Discussion section: “[…] recognition of the secondary structure of the stem-loop by MBLAC1 is important in maintaining the specificity of the cleavage” Why recognition? Is it not more likely that the structure prevents MBLAC1 to access the putative single-stranded cleavage sites?

We thank the editors for this useful comment. We have modified the text in agreement with the suggestion as follows: “The degradation patterns observed for RNA 4 and 5 suggest that the stem-loop substrate structure prevents MBLAC1 from accessing putative single-stranded cleavage sites located upstream of position 27, thus helping to maintain specificity of cleavage at position 27.”

5) The MBLAC1 RNA-ChIP experiment is not convincing. At a very minimum the authors should include an input sample to assay RDH expression relative to GAPDH and to confirm that RDH expression is not affected by the tagging of MBLAC1.

Following the suggestion, we have modified Figure 4 with a new graphical representation (Figure 4G) of the relative abundance of RD histone mRNA in both samples (wild-type and MBLAC1 FLAG knock in inputs). The relative abundance has been normalized to the abundance of GAPDH mRNA in the corresponding sample. The observed results are consistent with the RD histone mRNA level not being affected by the tagging of MBLAC1.

[Editors' note: further revisions were requested prior to acceptance, as described below.]

1) The interpretation of the shown plots is left as an exercise to the reader. To resolve this, 2 modifications can be suggested: 1) Indicate clearly in each plot the genomic region where an increased read-through upon knock-down of MBLAC1 and/or CPSF73 is observed, and 2) Sort or group plots into classes where read-through is observed, not observed or ambiguous. Once, this is done, please modify the text accordingly (it looks like only a subset of RD histone genes display read-through upon MBLAC1 knock-down).

We grouped the plots as you suggested based on the read-through effect (into major, possible, and not observed) and have colour-highlighted the genomic region where an increased read-through upon knock-down of MBLAC1 and/or CPSF73 was observed in each plot. We have also modified the main text (subsection “MBLAC1 is involved in histone-specific pre-mRNA processing in vivo”) and the figure accordingly.

2) The claimed gene-body normalization in panel A is not apparent from the shown data. Why?

See response to point 5 please.

3) The figure legend claims that regions (TSS-100 bp, TES+200 bp in panel A) and (TSS-200 bp, TES +200 bp in panel B) are depicted. Neither appears to reflect what is shown. Is it regions TES-100 to TES+200?

Thank you and apologies for the typo ‘TSS’ in the figure legend that now has been corrected to TES and the +/- from TES has been adjusted based on the now modified Figure 4*—*figure supplement 3.

4) Are the data showing the average of the 2 replicates or something else? This is not clarified.

The data represents the average of the 2 biological replicates. We are sorry for the lack of clarity. We have modified the text to make this clear.

5) Why should log2 transformation not be compatible with gene-body normalization of the data?

In the now modified Figure 4*—*figure supplement 3 the gene-body normalized data have been log 2 transformed to make the read-through region stand out more clearly – thank you for this suggestion. (The log 2 transformation was not applied to the previous version of the figure and this is probably the reason why the gene-body normalization was less appreciable in the previous version of the figure).